

# Spatiotemporal snow water storage uncertainty in the midlatitude American Cordillera

Yiwen Fang[1], Yufei Liu[2], Dongyue Li[1], Haorui Sun[1], Steven A. Margulis[1]

[1]Department of Civil and Environmental Engineering, University of California, Los Angeles, Los Angeles, 90095, USA
[2]China Institute of Water Resources and Hydropower Research, Beijing, 100048, China

*Correspondence to*: Steven A. Margulis (margulis@seas.ucla.edu)

**Abstract.** This work quantifies the uncertainty of accumulation-season peak snow water storage in the portions of the midlatitude American Cordillera where snow is a dominant driver of hydrology. This is accomplished through intercomparison of commonly used global and regional products over the Western U.S. (WUS) and Andes domains, which have similar

hydrometeorology but are disparate with respect to the amount of available in situ information. The recently developed WUS Snow Reanalysis (WUS-SR) and Andes Snow Reanalysis (Andes-SR) datasets, which have been extensively verified against in situ measurements, are used as baseline reference datasets in the intercomparison. Relative to WUS-SR climatological peak SWE storage (269 km$^3$), high- and moderate-resolution products (i.e. those with resolutions less than ~10 km) are in much better agreement (284 ± 14 km$^3$; overestimated by 6 %) compared to low-resolution products (127 km$^3$ ± 54 km$^3$;

underestimated by 53 %). In comparison to the Andes-SR peak snow storage (29 km$^3$), all other products show large uncertainty and bias (19 km$^3$ ± 16 km$^3$; underestimated by 34 %). Examination of spatial patterns related to orographic effects, showed that only the high- to moderate-resolution SNODAS and UA products show comparable estimates of windward-leeward SWE patterns over a subdomain (Sierra Nevada) of the WUS. Coarser products distribute too much snow on the leeward side in both the Sierra Nevada and Andes, missing orographic-rainshadow patterns that have important hydrological

implications. The uncertainty of peak seasonal snow storage is primarily explained by precipitation uncertainty in both the WUS ($R^2 = 0.55$) and Andes ($R^2 = 0.84$). Despite using similar forcing inputs, snow storage diverges significantly within the ERA5 (i.e. ERA5 vs. ERA5-Land) products and the GLDAS (modeled with Noah, VIC, and Catchment model) products due to resolution-induced elevation differences and/or differing model process representation related to rain–snow partitioning and accumulation-season snowmelt generation. The availability and use of in situ precipitation and snow measurements (i.e., in

WUS) in some products adds value by reducing snow storage uncertainty, however where such data are limited, i.e. in the Andes, significant biases and uncertainty exist.

## 1 Background and Motivation

Seasonal snow storage in mountains provide vital freshwater to downstream users estimated to be over 16.7 % of the global population (Immerzeel et al., 2020; Rhoades et al., 2022). Melt of accumulated winter snow in the spring and summer



impacts agriculture, hydropower generation, and water supply and recreation, making it a key component of the food-energy-water nexus in many regions of the world (Siirila-Woodburn et al., 2021; Huss et al., 2017; Qin et al., 2020). Despite its importance, a complete understanding of continental terrestrial water cycles is hampered by a limited characterization of seasonal mountain snow storage uncertainty.

The lack of in situ and remotely sensed measurements of mountain snow water equivalent (SWE), a key metric related
to water availability, are primarily responsible for the limited characterization of seasonal snow storage in these regions. For example, in the midlatitude American Cordillera, where snowmelt is estimated to contribute to as much as 70 % of total runoff in some basins (Li et al., 2017), existing in situ networks are both sparse and unrepresentative of the conditions spanning the larger domains in the Western United States (WUS) and South American Andes (Nolin et al., 2021; Dozier et al., 2016; Molotch and Bales, 2006, Saavedra et al., 2018). Current remotely sensed SWE estimates from passive microwave
measurements are useful over much of the globe, but are too coarse to capture the spatial heterogeneity and deep snowpacks in these regions with complex terrain (Luojus et al., 2021).

In lieu of measurements, globally available snow products, typically generated from land surface models (LSMs), provide the majority of large-scale estimates of the spatiotemporal patterns of mountain snow water storage. However, seasonal snow storage estimates from global snow products remains highly uncertain, which results from discrepancies in
meteorological forcings, variations in snow process representation, and coarse spatial resolution (Broxton et al., 2016b; Wrzesien et al., 2019; Cho et al., 2022; Liu et al., 2022). The uncertainty (including bias) of seasonal snow storage further propagates to streamflow forecasts (Kim et al., 2021) and impacts water resources management. Coarse spatial resolutions smooth topography and impact the ability to resolve orographic features (including rainshadows) over complex terrain (Daly, 2006). Current estimates of mountain snow water storage uncertainties in both space and time need to be characterized to
ensure the reliability of impact studies that rely on SWE estimates (e.g., Mankin et al., 2015; Immerzeel et al., 2020; Huning and AghaKouchak, 2020).

The analysis herein is applied to the snow-dominated midlatitude portions of the American Cordillera (Fig. 1), which are representative of regional mountains of significant importance to humans. To quantify the spatiotemporal uncertainties of snow storage from commonly used snow products, recently-developed high-resolution snow reanalysis datasets covering the
WUS (Fang et al., 2022) and Andes (Cortés and Margulis, 2017) are used in this work as reference datasets. The WUS and Andes domains have comparable atmospheric circulation patterns and hydrologic cycles (Rhoades et al., 2022), but are disparate with respect to the amount of available in situ information. The WUS has among the highest density of in situ snow information, which either directly or indirectly inform SWE estimates, while the Andes has little to no ground measurements, making SWE estimates almost entirely model-based. This paper aims to assess: 1) the spatiotemporal uncertainty of SWE in
the WUS and Andes over the accumulation season, and 2) the drivers of the SWE uncertainty. Knowledge of the uncertainty and its drivers will put current snow-impact studies in better context and provide a pathway for improving future estimates aimed at reducing SWE quantification uncertainty.





## 2 Study Domain and Datasets

### 2.1 Study Domain

This study focuses on the snow-dominated midlatitude mountain ranges of the America Cordillera (Fig. 1), where snowmelt-driven runoff serves large populations. Specifically, the WUS and Andes are selected as the study domains based on recently developed snow-specific reanalysis products (Fang et al., 2022; Cortés and Margulis, 2017). These SWE estimates have been significantly verified against independent in situ and airborne measurements, making them well-suited to being used as references for other products. The average elevation across the WUS is ~ 1383 m with a maximum > 4300 m, in contrast to

a higher average elevation of ~ 2999 m with a maximum > 6800 m in the Andes. The beginning of the seasonal snow cycle starts from 01 October and 01 April in the WUS and Andes, respectively. Hence, a water year (WY) spans from 01 October to 30 September in the WUS, and 01 April to 31 March in the Andes.

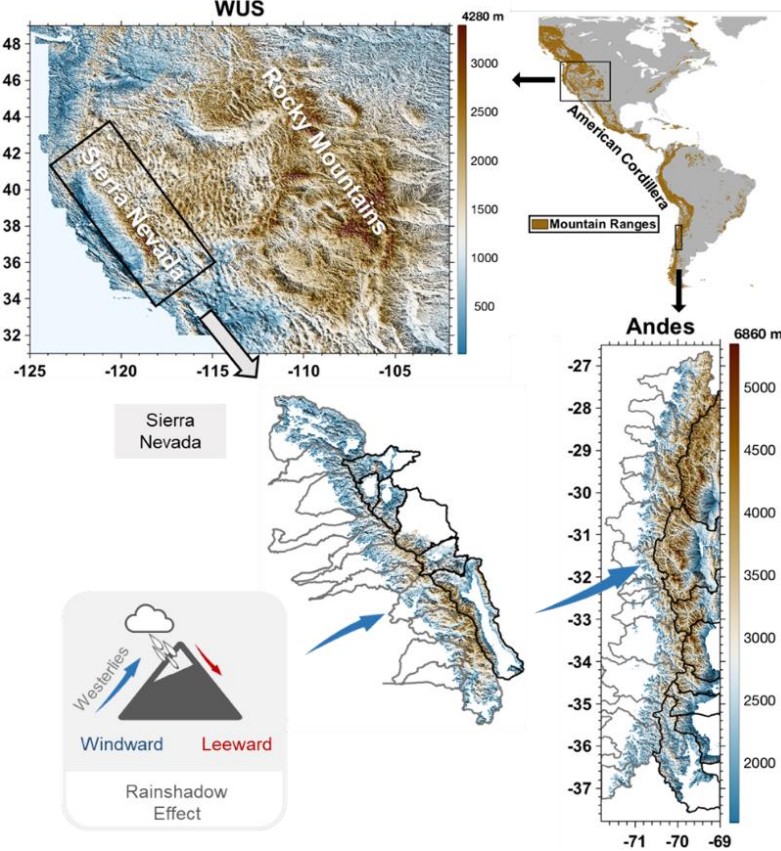

**Figure 1**. DEM and location of midlatitude American Cordillera including its subdomains in the Western U.S. (WUS) and

Andes. Bottom left cartoon highlights the typical rainshadow effect whereby shows that moist air rises on the windward side of a mountain depositing significant amounts of snow, and drier air flows down the leeward side of the mountain creating a rainshadow effect. Arrows represent the generalized directions of westerlies that drive orographic and rainshadow SWE patterns. The Sierra Nevada (SN, sub-basin of WUS) and Andes are chosen to study the rainshadow effect. Windward



watersheds are shown in gray boundaries and leeward watersheds are shown in black boundaries. Mountain ranges are based
on Snethlage et al. (2022).

The WUS contains three major mountain ranges including the Sierra Nevada, Rocky Mountains and Cascades (Figure
1). Amongst these, the Sierra Nevada subdomain is the closest analog to the Andes, sharing similar hydroclimatology and
topography. Winter westerlies dominate precipitation timing and patterns in these two mountain ranges, leading to orographic
gradients on the windward side of the mountains and rainshadow effects resulting in significant snow differences across
relatively short windward-leeward gradients.

## 2.2 Datasets

This paper intercompares data from the Andes Snow Reanalysis (Andes-SR) and WUS Snow Reanalysis (WUS-SR)
datasets (as reference datasets), to seven global snow datasets (available over both domains), and two regional datasets
(available only over the WUS domain) shown in Table 1. The Andes-SR (WYs 1985 to 2015; Cortés and Margulis, 2017)
SWE estimates are at ~ 180 m resolution and the WUS-SR (WYs 1985 to 2021; Fang et al., 2022) SWE estimates are at ~ 480
m resolution. The Andes-SR and WUS-SR datasets were both generated from the Bayesian framework developed by Margulis
et al. (2016, 2019) with assimilation of fractional snow-covered area images derived from Landsat 5, 7 and 8 using the Particle
Batch Smoother (PBS; Margulis et al., 2015). Independent verification shows that both datasets are consistent with in situ peak
SWE with a correlation coefficient of 0.73 over the Andes (Cortés and Margulis, 2017) and 0.77 (using > 25,000 station-years
of in situ data) over the WUS (Fang et al., 2022). Further verification of the WUS-SR SWE against Airborne Snow Observatory
(ASO) SWE estimates shows consistent performance between these two spatial products with correlation coefficients ranging
from 0.75 to 0.91. With high consistency against point-scale in situ and spatially-distributed airborne SWE estimates, as well
as the high spatial resolutions specifically targeting mountainous domains, these two snow reanalysis datasets are used as
reference SWE datasets to evaluate the snow storage of global and regional products over the WUS and Andes.

The seven global snow products include ERA5-Land, ERA5, MERRA2 and four GLDAS-2.1 products (GLDAS-
NOAH at 0.25°, GLDAS-NOAH at 1.0°, GLDAS-VIC at 1.0°, and GLDAS-CLSM at 1.0°). The SNODAS and UA products
only cover the US and therefore are not included in the Andes intercomparison. Following Liu et al. (2022), SWE, precipitation
and snowfall were collected from each of the seven global products, SNODAS, and UA (including PRISM precipitation (Daly
et al., 1994) used in the UA product). Since the reference snow reanalysis datasets do not output precipitation and snowfall,
only SWE is used for reference. For the purposes of analysis and discussion in this work, the products described above are
classified by their spatial resolution. Specifically, reference datasets and those products with spatial resolution less than ~1 km
are deemed "high-resolution" (*HR*: WUS-SR, Andes-SR, and SNODAS), those with spatial resolutions between ~1 km and
~10 km are deemed "moderate-resolution" (*MR*: UA, ERA5-Land), and those with spatial resolutions greater than ~10 km are
deemed "low-resolution" ("*LR*": ERA5, GLDAS subset). Globally and regionally available datasets are referred to as
"products" to distinguish them from the reference "datasets", i.e., WUS-SR and Andes-SR.





The snow reanalysis reference datasets are, by design, constrained by observations using a data assimilation approach. However, not all the products are solely model-based. SNODAS uses in situ snow, airborne SWE from gamma radiation snow surveys and satellite snow cover, and UA uses in situ SWE as inputs to constrain estimates. Although ERA5 assimilates snow

depth, limited examples of these in situ measurements are used in the WUS and Andes. However, in the WUS, with its relatively high density of in situ meteorological sites, almost all products are based on models with meteorological forcings that include some in situ measurements. In contrast, due to limited in situ meteorological sites in the Andes, the quality of input forcings remains unclear, but is likely more uncertain than over the WUS. More details on the snow products used herein are given in Table 1 and Appendix A.



**Table 1.** Details of snow datasets and products used in this work. Note that SNODAS data in WY 2004 is not used due to quality issues cited in NOHRSC (2004). *Snow reanalysis datasets are used as reference datasets.

| Datasets/Products | Land surface model | Spatial Resolution | Temporal Coverage | Forcings | Assimilated snow data (Method) | Domain availability |
|---|---|---|---|---|---|---|
| **WUS-SR*** | SSiB - SDC | 16" (~500 m) | 1985-2021 | MERRA2 | Landsat fSCA (PBS) | WUS |
| **ANDES-SR*** | SSiB - SDC | 6" (~180 m) | 1985-2015 | MERRA | Landsat fSCA (PBS) | Andes |
| SNODAS | NOHRSC Snow Model (NSM) | 1 km (~0.01°) | 2004-present | Downscaled NWP forcing | Ground based snow/ airborne SWE/ satellite snow cover (Newtonian Nudging) | WUS |
| UA | - | 4 km (~0.04°) | 1981-present | PRISM | In situ SWE/ snow depth from SNOTEL and snow depth from COOP (Ordinary Kriging Interpolation) | WUS |
| ERA5-Land | H-TESSEL (IFS Cy45r1) | 0.1° (~10 km) | | ERA5 with "lapse rate correction" | - | WUS, Andes |
| ERA5 | H-TESSEL (IFS Cy41r2) | 0.25° (~25 km) | 1950-present | IFS Cy41r2 with 4D-Var | In situ snow depth (Optimal Interpolation); IMS snow cover | WUS, Andes |
| MERRA2 | Catchment | 0.5° × 0.625° (~50 - 63 km) | 1980-present | MERRA2 | - | WUS, Andes |
| GLDAS-2.1 | Noah | 0.25° (~25 km) | | NOAA/GDAS, GPCP 1.3, bias corrected AGRMET | - | WUS, Andes |
| | Noah | 1° (~100 km) | 2001-present | | - | WUS, Andes |
| | VIC | 1° (~100 km) | | | - | WUS, Andes |
| | Catchment (CLSM) | 1° (~100 km) | | | - | WUS, Andes |





## 3 Intercomparison Methodology

### 3.1 Intercomparison study period

Where possible, the intercomparison study periods in the two domains are chosen as WYs 1985-2021 (01 October 1984 to 30 September 2021) for the WUS, and WYs 1985-2015 (01 April 1984 to 31 March 2015) for the Andes, based on the availability of the respective snow reanalysis datasets. Among the products listed in Table 1, only GLDAS (starting in WY 2001) and SNODAS (starting in WY 2005) are not available over the full snow reanalysis period. For those products, long-term climatologies are necessarily derived over the shorter periods. Hence in the WUS, climatologies for the GLDAS and

SNODAS products are over their available 21- and 17-year records, while all other products span the 37-year record. In the Andes, the GLDAS products are over their 15-year record, while all other products span the 31-year record. Analysis of climatological results from the products with longer periods do not show significant differences when applied to the shorter study periods (not shown).

### 3.2 Focusing on intercomparison during the snow accumulation season

The intercomparison herein focuses on the snow accumulation season. To motivate this focus, the climatological (long-term average) daily time series of domain-aggregated SWE volume across all products are illustrated in Fig. 2. Two key points are evident: (i) there are significant discrepancies between products (that are analyzed in more detail below) and (ii) much of the uncertainty occurs during the accumulation season (and then propagates to the ablation season). An accurate characterization of peak SWE (at the end of the accumulation season) is a key metric of the final condition of snow

accumulation processes and the initial condition leading into the main snowmelt season. Intercomparison of modeled snowmelt season processes are made more difficult when the initial conditions (i.e., peak SWE prior to the primary ablation season) across models are different. Given the large uncertainties observed in domain-wide peak SWE climatology (Fig. 2), this paper focuses on the uncertainties in the accumulation season (as done in Liu et al., 2022) in order to better understand how and why accumulation season estimates diverge across products. All the analyses focus on the accumulation season using metrics

described below.



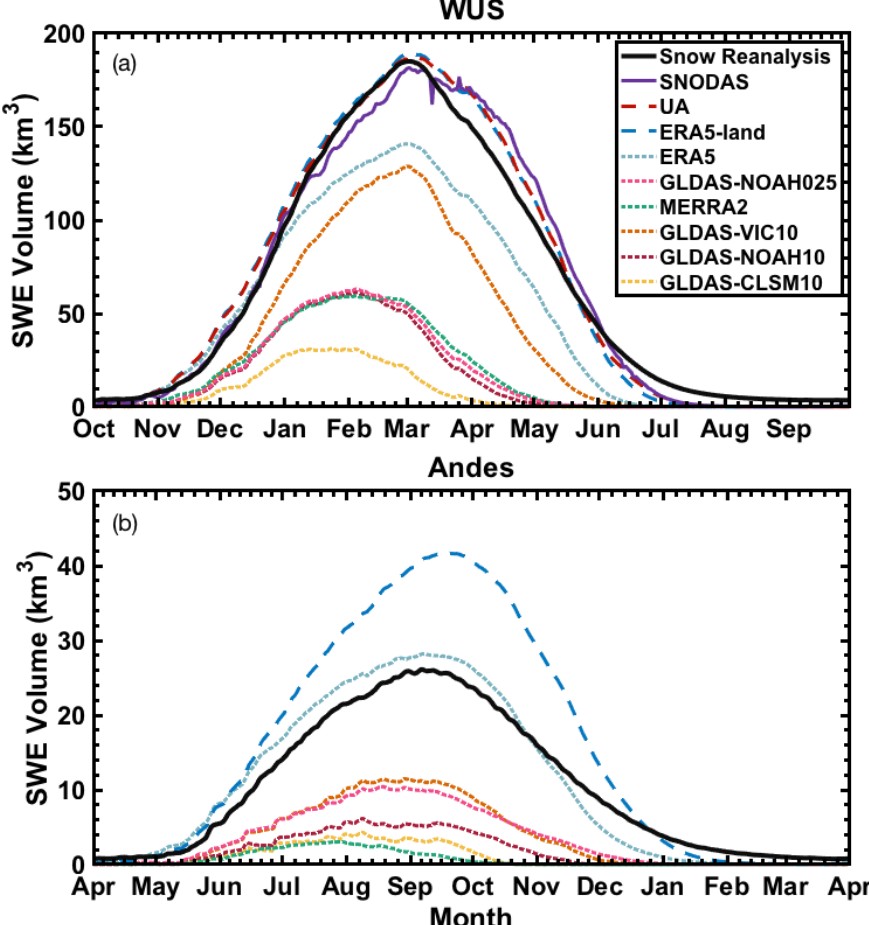

**Figure 2**. Climatology of seasonal cycle of SWE volume in the WUS and Andes domains. Solid lines represent high-resolution (HR) datasets and products, dashed lines represent moderate-resolution (MR) products, and dotted lines represent low-resolution (LR) products.

**3.3 Snow metrics used in the intercomparison**

The processes leading to the domain-aggregated peak SWE shown in Fig. 2 depend on pixel-scale snow mass balance processes. Hereafter, for each product, the pixel-wise processes are analyzed prior to aggregating to the larger domain. The day corresponding to pixel-wise peak SWE (defined as $t_{peak}$) is computed for each product at their raw spatial resolution. The pixel-wise peak SWE depth ($swe_{peak}$) is aggregated to get pixel-wise peak SWE volume ($SWE_{peak}$). Hence in results to follow,

$swe_{peak}$ is used to describe and analyze maps of SWE, while $SWE_{peak}$ is used to describe spatially aggregated volumes of SWE.

At each pixel, accumulation-season precipitation and snowfall are accumulated from the beginning of the WY up to $t_{peak}$, where the accumulated maps of SWE can then be aggregated over the domain of interest. The mass balance equations





relating domain-aggregated cumulative snowfall ($S_{acc}$), SWE ($SWE_{peak}$), cumulative ablation ($A_{acc}$), cumulative precipitation ($P_{acc}$), cumulative snowfall ($S_{acc}$), and cumulative rainfall ($R_{acc}$) are shown below:

$$SWE_{peak} = S_{acc} - A_{acc} \qquad (1)$$

$$S_{acc} = P_{acc} - R_{acc} \qquad (2)$$

where in Eq. (1) and (2), $P_{acc}$, $S_{acc}$, $SWE_{peak}$ are directly computed from the snow products. $R_{acc}$ and $A_{acc}$ are the residuals based on these two mass balance equations. Climatological values are computed as the long-term (interannual) mean of $P_{acc}$, $S_{acc}$, $SWE_{peak}$ over the intercomparison periods.

Persistent snow and ice areas are excluded before spatially integrating the SWE volumes, since most products analyzed in this work do not explicitly estimate glaciers and persistent snow. Such persistent snow and ice masks are first obtained from the Andes-SR and WUS-SR products and then aggregated to the spatial resolution of each product (as done in Liu et al., 2022). Domain masks in each product are also applied here, which are derived based on the reference datasets using the same approach. Details of persistent snow and ice masks and domain masks are described in Supplement S1, and shown
in Fig. S1 and S2.

    Beyond domain-wide results, we choose to intercompare products and their ability to capture rainshadow effects, that often occur over short geographic scales, but have significant influence on the water availability between windward and leeward sides of mountain ranges. For simplicity we focus on the windward-leeward contrasts over the Sierra Nevada in the WUS and those over the Andes. Figure 1 shows the boundaries of windward basins (in gray) and leeward basins (in black) for
both domains. The Sierra Nevada and Andes are analogs of each other due to the mostly north-south orientation of the mountain ranges that are relatively perpendicular to the mostly westerly prevailing winds. In both cases, the windward and leeward basins serve distinct downstream populations and so resolving those spatial variations have important hydrological implications. To assess the ability of products in capturing rainshadow effects, pixel-wise $SWE_{peak}$ is aggregated over the windward ($SWE_{peak}^{wind}$) and leeward ($SWE_{peak}^{lee}$) watersheds. Since MR and CR pixels may cover both windward and leeward
watersheds, fractional $swe_{peak}$ is aggregated to get $SWE_{peak}$ over the two types of watersheds separately (Fig. S3 and S4). The fractional $swe_{peak}$ is computed by multiplying pixel-wise $swe_{peak}$ and the fraction of pixel within the windward or leeward watershed. The detailed steps used to derive the windward and leeward watershed snow storage are described in S2.



## 4 Results and Discussion

### 4.1 Climatological SWE uncertainty

#### 4.1.1 Spatial distribution of pixel-wise peak SWE

Climatological pixel-wise $swe_{peak}$ maps for the WUS-SR (Fig. 3a) clearly show the highest snow storage occurring in the Sierra Nevada, Cascades, and Rocky Mountains. When integrated over the whole domain, the climatological WUS $SWE_{peak}$ is 269 km$^3$ (Fig. 3k). Similar spatial distributions of $swe_{peak}$ are observed for the HR (SNODAS; Fig. 3b) and MR products (UA and ERA5-Land; Fig. 3c and 3d). However, the remaining products (ERA5, MERRA5 and GLDAS subset; Fig. 3e to 3j)

significantly underestimate $swe_{peak}$ and smooth out the spatial patterns captured by the HR and MR products. The combined HR and MR inter-product average of climatological WUS $SWE_{peak}$ is 284 ± 14 km$^3$, in contrast to an average of 127 ± 54 km$^3$ for LR products (Fig. 3k). This suggests large uncertainty (both bias and spread) in $SWE_{peak}$ among LR products. Compared to WUS-SR, SNODAS overestimates $SWE_{peak}$ by ~12 % (Fig. 3k) and exhibits higher $swe_{peak}$ in the Sierra Nevada, Cascades, and Rocky Mountains. UA and ERA5-Land both exhibit a similar magnitude of $SWE_{peak}$ (differences < 5 %) compared to

WUS-SR, both of which have higher $swe_{peak}$ in the Cascades. Despite similar spatial distribution of $swe_{peak}$, ERA5 underestimates WUS $SWE_{peak}$ by 22 % (Fig. 3k) compared to WUS-SR. All GLDAS products severely underestimate $SWE_{peak}$, where GLDAS-VIC10 shows the highest WUS $SWE_{peak}$ (with a 35 % underestimation compared to WUS-SR).

Based on the Andes-SR, the climatological $SWE_{peak}$ is 29 km$^3$ (Fig. 4i). The southern Andes has higher $swe_{peak}$ compared to the northern region (Fig. 4a). The spatial distribution of $swe_{peak}$ and integrated $SWE_{peak}$ volumes vary much more

broadly across different products (Fig. 4b to 4k) than they do in the WUS. The MR and LR inter-product average of climatological $SWE_{peak}$ is 19 ± 16 km$^3$ (Fig. 4i). ERA5-Land and ERA5 overestimate $SWE_{peak}$ by 66 % and 18 %, respectively (Fig. 4i). ERA5-Land significantly overestimates $swe_{peak}$ in the southern part of the Andes. Most of the LR products, including MERRA2 and the GLDAS subset, significantly underestimate $SWE_{peak}$ by as much as 79 % (MERRA2), compared to Andes-SR (Fig. 4i). These findings for the Andes domain are qualitatively similar to Liu et al. (2022), where ERA5 and ERA5-Land

overestimate $SWE_{peak}$ and MERRA2 and GLDAS underestimate $SWE_{peak}$ in High Mountain Asia (HMA), another snow-dominated region with limited in situ measurements.



**Figure 3**. (a – j) Spatial distribution of climatological swe$_{peak}$ in the WUS. (k) shows the climatological WUS SWE$_{peak}$ (colored bars) and the interannual inter-quartile range (IQR; black error bars). The bar plots are ordered by spatial resolution, with highest resolution on the left and lowest resolution on the right. The vertical dashed lines separate the three spatial resolution categories (i.e., HR < ~ 1 km, ~ 1 km < MR < ~ 10 km, LR > ~ 10 km). Glacier and permanent snow areas are masked out in the maps and domain aggregated volumes.







**Figure 4**. (a – h) Spatial distribution of climatological swe$_{peak}$ in the Andes. (i) shows the climatological Andes SWE$_{peak}$
(colored bars) and the interannual inter-quartile range (IQR; black error bars). The bar plots are ordered by spatial resolution,
with highest resolution on the left and lowest resolution on the right. The vertical dashed lines separate the three spatial
resolution categories (i.e., HR < ~ 1 km, ~ 1 km < MR < ~ 10 km, LR > ~ 10 km). Glacier and permanent snow areas are
masked out in the maps and domain aggregated volumes.

### 4.1.2 Resolving key spatial gradients: Rainshadow effects

220        In addition to the overall spatial distribution in SWE, the orographically-driven rainshadow (windward vs. leeward)
distribution represents an example of an important spatial feature in many mountain contexts. In mountain ranges exposed to



persistent prevailing winds, it is expected that the windward side of the range will have more SWE than the leeward side (Fig. 1). While significant biases exist in products as described in Section 4.1.1, this section focuses on the relative patterns of windward vs. leeward storage. The differences in rainshadow storage gradients are specifically examined in the Sierra Nevada

subdomain of the WUS and the Andes. While resolving rainshadow effects is challenging for narrow topographic regions like the Sierra Nevada and Andes, it has important hydrological implications as large gradients in SWE storage propagate to spring/summer runoff and streamflow that supply downstream users.

Based on the WUS-SR in the Sierra Nevada (Fig. 5), the latitudinal distribution of $SWE_{peak}^{wind}$ is the largest in the 37°–38° N latitudinal band, while the latitudinal distribution of $SWE_{peak}^{lee}$ is the largest in the 38°–39° N latitudinal band. The

latitudinal windward and leeward storage of SWE decreases monotonically north and south of these maximum values. The total stored windward volume $SWE_{peak}^{wind}$ is 3.74 times more than the leeward volume $SWE_{peak}^{lee}$. This ratio is the combined effect of variations in area and SWE depth between the windward and leeward basins and identifies that (on average) the windward basins store between 3 and 4 times more SWE volume than the leeward basins. Given that the windward and leeward areas across which SWE is integrated are effectively the same across products, any differences in windward-leeward ratio are driven

by differences in SWE depth. SWE depth variations are primarily driven by resolving orographic enhancement of snowfall between windward and leeward slopes. In the Sierra Nevada, only SNODAS and UA products (spatial resolutions < ~ 4 km) exhibit comparable $SWE_{peak}^{wind}$ to $SWE_{peak}^{lee}$ ratios. The ratios of $SWE_{peak}^{wind}$ to $SWE_{peak}^{lee}$ are 4.20 (12 % greater than the WUS-SR) for SNODAS and 3.14 (16 % less than WUS-SR) for UA, suggesting a fairly good agreement between windward-leeward snow volume distributions in these products. However, resolving the pattern of windward-leeward snow distribution is

significantly impaired in the other MR and LR products. The ratios computed from ERA5-Land, ERA5, GLDAS-NOAH025, MERRA2, GLDAS-VIC10, GLDAS-NOAH10, and GLDAS-CLSM10 range from 1.08-2.40 and are 36 %, 46 %, 43 %, 55 %, 68 %, 66 %, and 71 % less than that in the WUS-SR, respectively. Hence, the MR and LR products generally have too little snow on the windward side compared to the leeward side. The location of the windward maximum $SWE_{peak}$ is consistent in most snow products with the exception of the LR products (i.e., ERA5, GLDAS), which have a secondary maximum between

39°–40° N. The location of the leeward maximum $SWE_{peak}$ is consistent in most of the snow products with the exception of MERRA2, which is maximum at a lower latitude.

Based on the Andes-SR, the largest $SWE_{peak}^{wind}$ is distributed in the 35°–36° S latitudinal band, while the distribution of $SWE_{peak}^{lee}$ has two local maxima in the 31°–32° S and 35°–36° S latitudinal bands (Fig. 6). The ratio of $SWE_{peak}^{wind}$ to $SWE_{peak}^{lee}$ is 1.58 from the Andes-SR, which is again the combined effect of windward-leeward variations in both area and SWE depth.

Like the Sierra Nevada, ERA5-Land and all of the CR products improperly partition $SWE_{peak}$ over the windward vs. leeward basins in the Andes. These products have $SWE_{peak}^{wind}$ to $SWE_{peak}^{lee}$ ratios less than 1 indicating deficient snow in the windward watersheds compared to the leeward watersheds. The lowest $SWE_{peak}^{wind}$ to $SWE_{peak}^{lee}$ ratio of 0.72 is observed from MERRA2 (54 % less than Andes-SR). GLDAS-VIC10 has the largest $SWE_{peak}^{wind}$ to $SWE_{peak}^{lee}$ ratio of 0.92 among Andes global products,





which is still 42 % less than the Andes-SR. For the windward watersheds, $SWE_{peak}^{wind}$ from ERA5-Land and GLDAS-CLSM10

are the highest in the same latitudinal band as the Andes-SR, however, the other products have an erroneous $SWE_{peak}^{wind}$

distribution. None of the products resolve the $SWE_{peak}^{lee}$ distribution on the leeward side.

**Figure 5**. Latitudinal distribution of integrated $SWE_{peak}$ (km$^3$) over windward ($SWE_{peak}^{wind}$; light gray areas) and leeward basins

($SWE_{peak}^{lee}$; dark gray areas) in the Sierra Nevada in first and third columns. Text labels indicate the ratio of latitudinally-

integrated $SWE_{peak}^{wind}$ to $SWE_{peak}^{lee}$. The climatological $swe_{peak}$ (m) spatial patterns corresponding to the latitude bands indicated

by dashed boxes are illustrated in the second and fourth columns. The red line represents the Sierra Nevada ridgeline separating



windward (western) from leeward (eastern) basins. Note: Different $swe_{peak}$ ranges are used for each product to highlight latitudinal/spatial patterns more than absolute values (due to significant biases in some products).

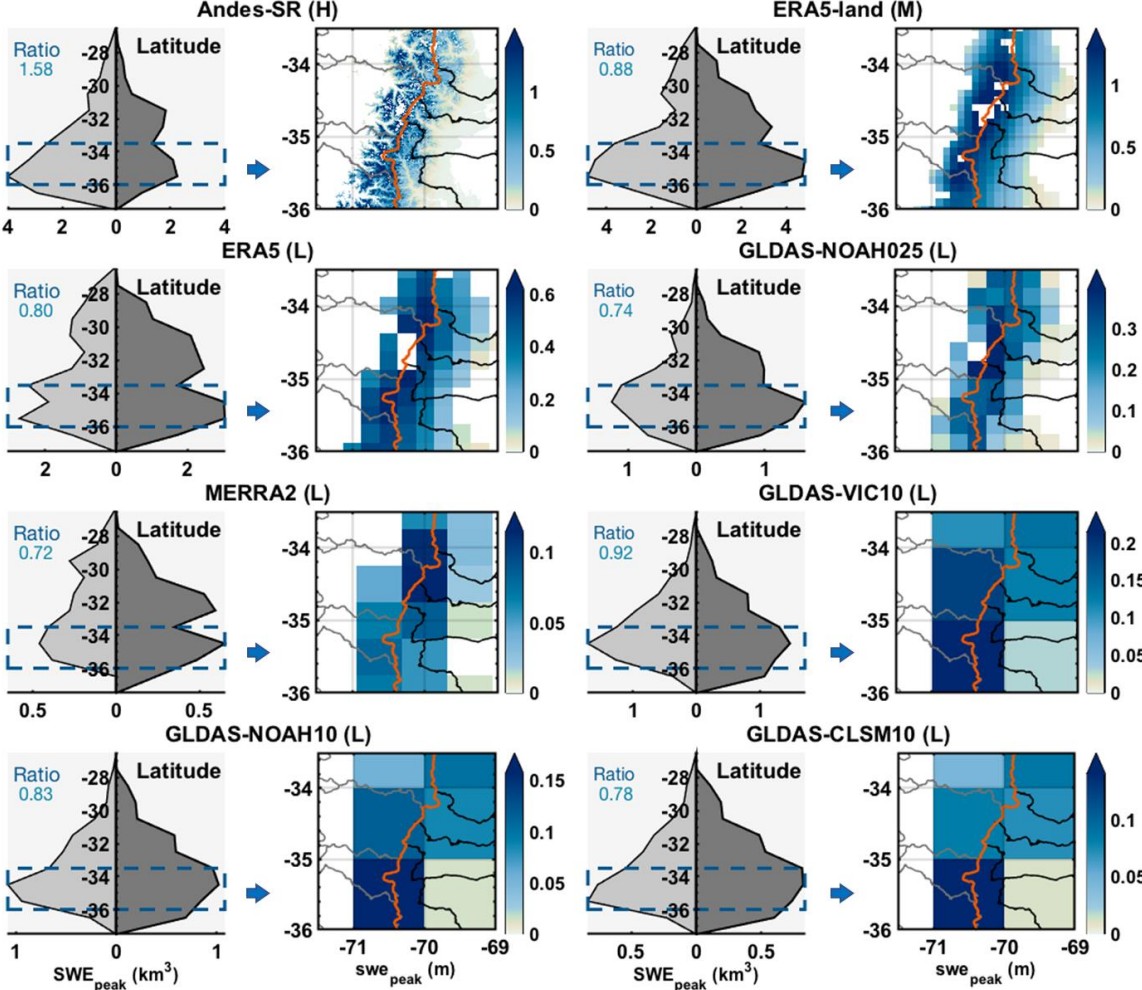

**Figure 6**. Latitudinal distribution of $SWE_{peak}$ (km³) over windward ($SWE_{peak}^{wind}$; light gray areas) and leeward basins ($SWE_{peak}^{lee}$; dark gray areas) in the Andes in first and third columns. Text labels indicate the ratios of latitudinally-integrated $SWE_{peak}^{wind}$ to $SWE_{peak}^{lee}$. The climatological $swe_{peak}$ (m) spatial patterns corresponding to the latitude bands indicated by dashed boxes are illustrated in the second and fourth columns. The red line represents the Andes ridgeline separating windward (western) from leeward (eastern) basins. Note: Different $swe_{peak}$ ranges are used for each product to highlight latitudinal/spatial patterns.

## 4.2 Interannual SWE uncertainty

The interannual variability of $SWE_{peak}$ is in general agreement (with correlation coefficients $R > 0.85$) between the WUS-SR snow reanalysis and other products shown in Fig. 7. $SWE_{peak}$ from UA and ERA5-Land agrees well with WUS-SR



in both magnitude and correlation (Fig. 7b and 7c), with relative mean differences (*RMD*) of less than 3 % in absolute value and *R* > 0.9. While SNODAS overestimates SWE volume with a *RMD* of 14 % (Fig. 7a), it shows consistent interannual

variations with a high *R* value of 0.92. The LR products are generally well correlated with WUS-SR, although $SWE_{peak}$ from these products is underestimated by as much as 190 km$^3$ (GLDAS-CLSM10), equivalent to a *RMD* of 71% compared to WUS-SR. Figure 7j shows that SWE percentiles computed from different products in the WUS are in better agreement in extreme years and in less agreement for near-average years. For example, WY 2017 was the wettest year among all products and WY 2015 was the driest year for all products except for SNODAS (in which WY 2005 is suspiciously low). WY 2014 was a

normal-to-wet year with $SWE_{peak}$ between the sixtieth and seventieth percentiles from GLDAS-NOAH025, MERRA2, GLDAS-NOAH10, and GLDAS-CLSM10, but a normal-to-dry year with $SWE_{peak}$ less than the fiftieth percentile in the other products.

The interannual variability of $SWE_{peak}$ is in much less agreement in the Andes (Fig. 8; with *R* as low as 0.56). Fig. 8 shows that ERA5-Land and MERRA2 are most consistent with Andes-SR in terms interannual variability (*R* > 0.85). However,

ERA5-Land overestimates $SWE_{peak}$ by 18 km$^3$ (*RMD* = 65 %) and MERRA2 underestimates $SWE_{peak}$ by 23 km$^3$ (*RMD* = -80 %). Although ERA5 has the smallest *RMD* of 17 %, the correlation coefficient *R* is 0.74, suggesting that $SWE_{peak}$ from ERA5 is less representative of interannual variation in the Andes. For the GLDAS products, GLDAS-NOAH025 has *R* = 0.79, whereas *R* values for other GLDAS products at 1° are less than 0.65, indicating that SWE from these LR products are less consistent with the interannual variation from Andes-SR. Figure 8a illustrates that the $SWE_{peak}$ percentiles computed from the

common 12-year record are much less consistent in the Andes than in the WUS (shown in Fig. 7j) for both normal and extreme years. Despite good temporal correlation of $SWE_{peak}$ (R > 0.86) in WUS, the relatively poorer temporal correlations (R > 0.56) identified from the LR products in the Andes, indicate that they may be less suitable for trend or other analyses that require snow estimates with representative interannual variability.



**Figure 7**. Scatter plots (a – i) of SWE$_{peak}$ volumes between WUS-SR and other products. Each dot represents SWE$_{peak}$ volume (km$^3$) for each year over the study period (WYs 1985 to 2021) where data is available. For the SNODAS and GLDAS products, the comparison is over WYs 2005 to 2021, and 2001 to 2021, respectively. (j) shows the SWE$_{peak}$ percentiles in each WY over the overlapping period including all products (WYs 2005 to 2021).



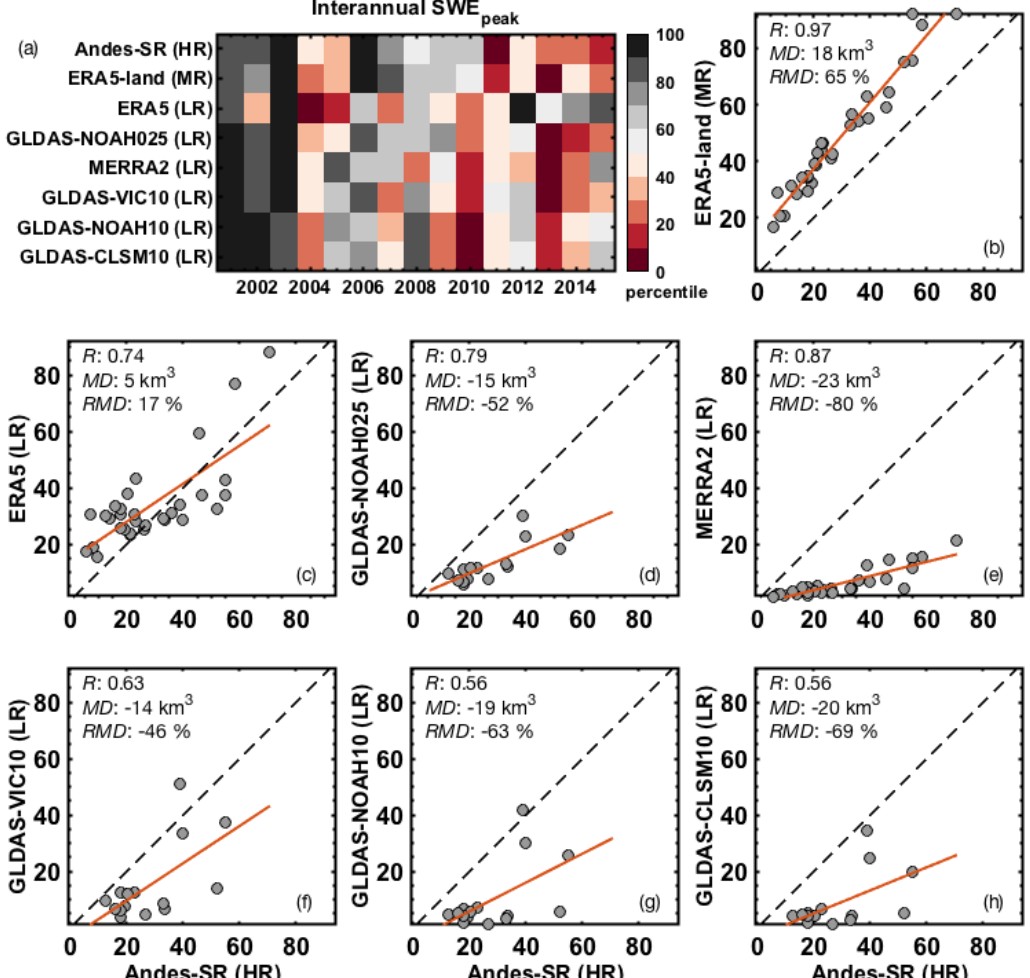

**Figure 8**. Scatter plots (b – h) of SWE$_{peak}$ volumes between Andes-SR and other products. Each dot represents SWE$_{peak}$ volume (km$^3$) for each year over the study period (WYs 1985 to 2015) where data is available. For the GLDAS products, the comparison is over WYs 2001 to 2015. (a) shows the SWE$_{peak}$ percentiles in each WY over the overlapping period including all products (WYs 2001 to 2015).

### 4.3 Drivers of SWE uncertainty

### 4.3.1 Uncertainty in annual SWE$_{peak}$ primarily explained by accumulation-season precipitation and snowfall

To better understand the accumulation-season SWE$_{peak}$ uncertainty driven by model inputs, the relationship among P$_{acc}$, S$_{acc}$, and SWE$_{peak}$ is quantified for all products. The annual data points are more clustered in the WUS (Fig. 9a, b) than those in the Andes (Fig. 9c, d). GLDAS-CLSM10 and MERRA2 tend to have lower P$_{acc}$, S$_{acc}$ and therefore SWE$_{peak}$ in both the WUS and Andes. ERA5-Land and ERA5, on the other hand, have higher P$_{acc}$, S$_{acc}$ and SWE$_{peak}$ in both domains. For rain-




snow partitioning, UA (Fig. 9a) tends to have more $S_{acc}$ over the WUS compared to the other products. Given similar $S_{acc}$, SNODAS (Fig. 9b) is inclined to generate higher $SWE_{peak}$. In the Andes, GLDAS-NOAH10 and GLDAS-CLSM10 partition less $P_{acc}$ into $S_{acc}$ (circles lower than the regression line), in contrast to ERA5-Land and ERA5 that tend to partition more (Fig. 9c). $SWE_{peak}$ from ERA5-Land diverges from ERA5 (Fig. 9d) given similar amount of $P_{acc}$ and $S_{acc}$, presumably caused by different melt amounts between the two products driven by resolution-induced elevation differences.

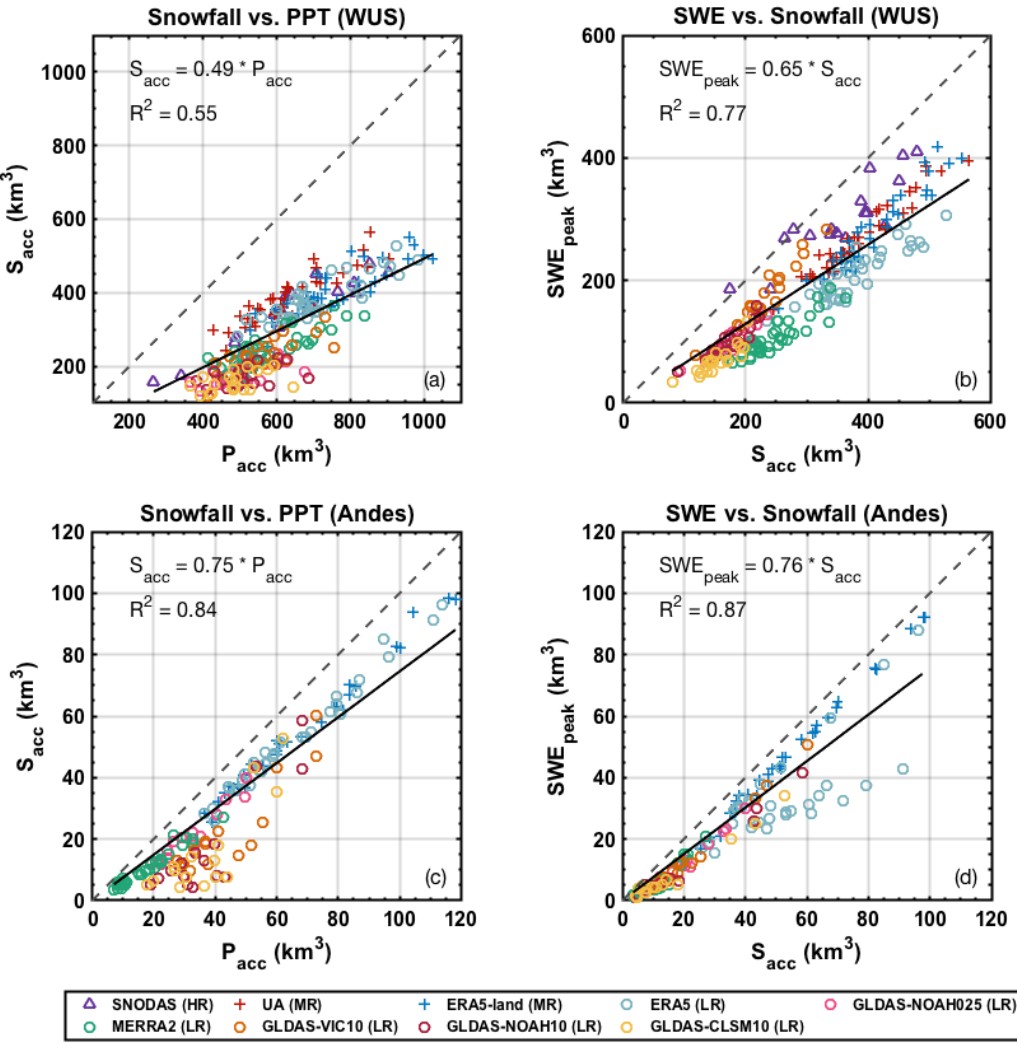


**Figure 9**. Left panels show scatter plots of accumulation-season $S_{acc}$ (km$^3$) vs. $P_{acc}$ (km$^3$) volumes over the WUS and Andes, respectively, indicating the partitioning of precipitation into snowfall. Right panels show scatter plots of accumulation-season $SWE_{peak}$ (km$^3$) vs. $S_{acc}$ (km$^3$) over WUS and Andes, respectively, indicating how much snowfall remains as SWE vs. being lost to ablation. Solid lines are linear regression and dashed lines are 1:1 lines.



Annual values of $P_{acc}$ and $S_{acc}$ estimates from all products show that the variance in $P_{acc}$ explains the majority of the variance in snowfall in the accumulation season with a coefficient of determination $R^2 = 0.55$ in the WUS (Fig. 9a) and $R^2 = 0.84$ in the Andes (Fig. 9c). This is consistent with previous findings (Cho et al., 2022; Broxton et al., 2016b; Liu et al., 2022) and the expectation that precipitation is the major contributor to uncertainy in SWE. The lower $R^2$ in the WUS compared to the Andes suggests that other factors such as air temperature plays a more important role in rain–snow partitioning in the WUS.

Approximately 49 % of $P_{acc}$ falls as snow in the WUS, whereas, around 75 % of $P_{acc}$ falls as snow in the Andes (Fig. 9a, c). This is because the Andes is at higher average elevation (~2999 m) with cooler temperature than the WUS (~1383 m), leading to more precipitation falling as snow. The variance in $SWE_{peak}$ is mostly explained by the variance in $S_{acc}$, i.e., $R^2 = 0.77$ in WUS (Fig. 9b) and $R^2 = 0.87$ in the Andes (Fig. 9d). As a fraction of cumulative snowfall, 65 % and 76 % remains as $SWE_{peak}$ in the WUS and Andes, respectively, while the rest is lost to accumulation-season ablation.

**4.3.2 Uncertainty in climatological $SWE_{peak}$ significantly impacted by differences in LSMs and spatial resolution**

To understand the impact of varying LSM mechanisms (i.e., rain–snow partitioning and snowmelt generation) and spatial resolution on the uncertainties in SWE, the climatological precipitation, snowfall, and $SWE_{peak}$ for all products over the WUS and Andes are shown in Fig. 10. The rainfall to precipitation ratio ($R_{acc}/P_{acc}$, gray text) represents the impact of rain–snow partitioning mechanisms, and the ablation to snowfall ratio ($A_{acc}/S_{acc}$, black text) represents the impact of accumulation-

season snowmelt mechanisms. It should be noted that different peak SWE days may impact $R_{acc}/P_{acc}$ via the accumulation window, i.e., the shorter accumulation season in the GLDAS subset (associated with earlier peak SWE days, $t_{peak}$, Fig. 10 red symbol) has cooler average temperature, and thus lower $R_{acc}/P_{acc}$. However, no significant relationship was found between $t_{peak}$ and $R_{acc}/P_{acc}$, suggesting that $R_{acc}/P_{acc}$ is not sensitive to $t_{peak}$. The WUS, with relatively lower elevation, has higher precipitation in the form of rainfall and higher snowfall loss to ablation than the Andes at higher elevation. In the WUS, $R_{acc}/P_{acc}$ ranges

from 0.39 (UA) to 0.69 (GLDAS-CLSM10), and $A_{acc}/S_{acc}$ ranges from 0.15 (SNODAS) to 0.56 (MERRA2). In the Andes, $R_{acc}/P_{acc}$ ranges from 0.19 (ERA5-Land) to 0.57 (GLDAS-CLSM10), and $A_{acc}/S_{acc}$ ranges from 0.13 (ERA5-Land) to 0.48 (GLDAS-CLSM10). Precipitation tends to fall more as snow in the HR, MR, and ERA5 products, whereas a higher fraction of precipitation falls as rainfall in the other products (GLDAS, MERRA2), even though lower $P_{acc}$ are observed in both domains. The differences in melt mechanisms across product models further differentiate the $A_{acc}/S_{acc}$, and therefore $SWE_{peak}$.





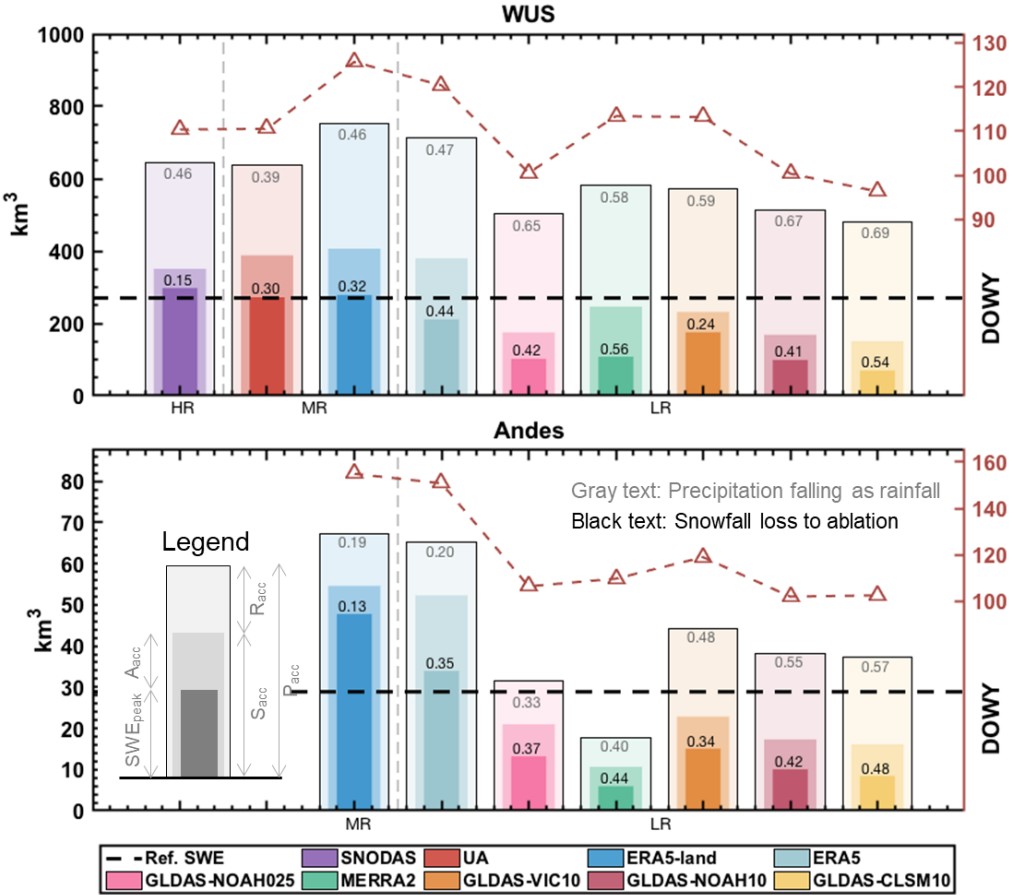


**Figure 10**. Climatological $SWE_{peak}$, $S_{acc}$, and $P_{acc}$ volumes aggregated over WUS (top panel) and Andes (bottom panel) in $km^3$. Red triangles (corresponding to right y-axis) shows the $t_{peak}$ averaged over all pixels and WYs. The horizontal dashed lines are the reference snow reanalysis SWE volumes from WUS-SR and Andes-SR. The vertical dashed lines group the products by spatial resolution (i.e., HR, MR, LR). The black text lists the $A_{acc}/S_{acc}$ and gray text lists the $R_{acc}/P_{acc}$.

Accumulation-season snowfall and $SWE_{peak}$ are sensitive to different rain–snow partitioning and snowmelt generation mechanisms across products. The same precipitation inputs (with only minor differences caused by downscaling) are used to derive GLDAS estimates at 1.0° from three different LSMs, making the GLDAS models a useful subset to understand the impact of LSM process representation on SWE estimates. Among the GLDAS subset at 1.0°, $R_{acc}/P_{acc}$ and $A_{acc}/S_{acc}$ range from 0.59-0.69 and 0.24-0.54, respectively, in the WUS (Fig. 11a), and range from 0.48-0.57 and 0.34-0.48, respectively, in the

Andes (Fig. 11b). Compared to $R_{acc}/P_{acc}$, a wider range of $A_{acc}/S_{acc}$ values are observed in both the WUS and Andes, suggesting that snowmelt generation mechanism differences contribute more to the climatological $SWE_{peak}$ uncertainties than the rain-snow partitioning differences. Given a similar amount of $P_{acc}$, GLDAS-VIC10 partitions the most into snowfall even with later peak days, whereas GLDAS-CLSM10 partitions the least in both domains. The differences in $R_{acc}/P_{acc}$ are ≤ 0.1 between



GLDAS-VIC10 and GLDAS-CLSM10, implying that the differences in snowfall caused by rain–snow partitioning is less than
10% of precipitation inputs among the GLDAS subset at 1.0°. For $A_{acc}/S_{acc}$, GLDAS-VIC10 has the lowest ratio compared to
others, suggesting that VIC snowfall loss to ablation is the least in this domain. GLDAS-VIC10 tends to have higher $P_{acc}$, $S_{acc}$,
and $SWE_{peak}$ which are closer to those from the HR or MR snow products. In the VIC model, elevation bands are used to better
represent sub-grid snowfall and SWE estimates. In addition to GLDAS-CLSM10, which has the highest $A_{acc}/S_{acc}$ among the
GLDAS subset at 1.0°, MERRA2 has a comparably high $A_{acc}/S_{acc}$. Both products use the same LSM, suggesting that a larger
portion of snowfall is lost as accumulation-season ablation in the Catchment model (Xiao et al., 2021).

Domains with larger variance in elevation are likely to be more sensitive to model spatial resolution, and therefore
impact elevation-dependent mechanisms in the LSMs. ERA5-Land (0.1°) and ERA5 (0.25°) SWE are derived from the same
LSM driven by similar forcings but modeled at different spatial resolutions. Similarly, GLDAS-NOAH025 (0.25°) and
GLDAS-NOAH10 (1°) SWE are derived from the same Noah model driven by similar forcings but at two spatial resolutions.
These two groups of products (Fig. 11c to f) are useful to isolate the impact of spatial resolution on SWE estimates via
differences in elevation representation. The raw DEMs from each product are used to compute the mean and standard deviation
of elevation over WUS and Andes. The Andes, located at higher elevation also has a larger variance in elevation (standard
deviation > 1100 m) compared to the WUS (standard deviation < 800 m) for any resolution. The standard deviation varies
more significantly with resolution than the mean in both WUS and Andes (Fig. 11g, h). With coarser spatial resolution, the
variance in elevation decreases, indicating that coarse-resolution products tend to underestimate the true variance in elevation.
The differences in elevation variance between products are larger in the Andes than the WUS. For example, when increasing
resolution of GLDAS from 1.0° (~ 100 km) to 0.25° (~ 25 km), the standard deviation of elevation increases by 14% in the
Andes compared to 8% in the WUS. The $R_{acc}/P_{acc}$ is similar in the ERA5-Land and ERA5 for the same domains (i.e., 0.46
from ERA5-Land and 0.47 from ERA5 in the WUS; 0.19 from ERA5-Land and 0.20 from ERA5 in the Andes), suggesting
that the rain–snow partitioning in the ERA5 models is relatively insensitive to the elevation differences introduced by different
spatial resolutions. The Andes is located at a higher elevation than the WUS, resulting in lower $R_{acc}/P_{acc}$. However, the $A_{acc}/S_{acc}$
varies significantly between ERA5-Land and ERA5 in both WUS (0.32 vs. 0.44, respectively) and Andes (0.13 vs. 0.35,
respectively). Hence, even though similar amounts of snowfall are generated for ERA5 and ERA5-Land, $SWE_{peak}$ can be
significantly different due to differences in ablation resulting from spatial resolution-based elevation differences. For GLDAS-
NOAH025 and GLDAS-NOAH10, $R_{acc}/P_{acc}$ and $A_{acc}/S_{acc}$ are similar in the WUS. Large differences of both $R_{acc}/P_{acc}$ and
$A_{acc}/S_{acc}$ are observed between GLDAS-NOAH025 and GLDAS-NOAH10. This suggests that increasing spatial resolution
from 0.25° to 0.1° (ERA5 subsets) significantly impact snowmelt generation in both Andes and WUS, whereas increasing
spatial resolution from to 1° to 0.25° (GLDAS subsets) impacts rain-snow partition and snowmelt generation only in Andes
with its larger differences in standard deviation of elevation between products at two different spatial resolutions.





**Figure 11**. (a) and (b) $R_{acc}/P_{acc}$ and $A_{acc}/S_{acc}$ for three GLDAS LSMs (VIC, Noah, and Catchment) at the same spatial resolution (~ 100 km). (c) and (d) $R_{acc}/P_{acc}$ and $A_{acc}/S_{acc}$ for ERA5-Land (~ 10 km) and ERA5 (~ 25 km) using the same LSM and similar forcings, but different spatial resolutions. (e) and (f) $R_{acc}/P_{acc}$ and $A_{acc}/S_{acc}$ for GLDAS-NOAH025 (~ 25 km) and GLDAS-NOAH10 (~ 100 km) using the same LSM and similar forcings, but different spatial resolutions. (g) and (h) mean and standard deviation of elevation over WUS and Andes from the ERA5-Land and ERA5 group, and the GLDAS-NOAH025 and GLDAS-NOAH10 group (where colors represent the products shown in (c)-(f)).





## 5 Conclusion

This paper quantifies the spatiotemporal snow storage uncertainty over the midlatitude American Cordillera (i.e., the intermountain WUS and Andes) that is influenced significantly by snow processes. These two domains are both snow-
dominated areas sharing similar hydrometeorology, however, much less in situ measurements are available in the Andes compared with the WUS. The uncertainties of snow water storage, spatial patterns (including orographic-rainshadow effect), and interannual variability are analyzed among the high-resolution (HR, less than ~1 km), moderate-resolution (MR, between ~1 km to ~10 km) and low-resolution (LR, greater than ~10 km) snow products. The impact of forcings, LSM differences and spatial resolution on snow storage uncertainty is assessed to provide insights for generating future snow products especially
for snow-dominated regions including areas with scarce in situ measurements.

With respect to characterizing climatological and interannual storage uncertainty, the key conclusions are:

1) In the WUS, HR and MR snow products are in better agreement with WUS-SR peak snow storage (269 km$^3$) than the LR snow products, among which snow storage is biased low with large uncertainty. The climatological snow storage was found to be 284 km$^3$ ± 14 km$^3$ among HR and MR products and 127 km$^3$ ± 54 km$^3$ among LR
products. In the Andes, MR and LR products exhibit much larger relative uncertainty in snow storage. The Andes-wide peak snow storage estimates are less clustered by spatial resolution with climatological estimates of 19 km$^3$ ± 16 km$^3$ compared with peak snow storage of 29 km$^3$ from Andes-SR.

2) Beyond significant biases in overall storage, most of the global products poorly characterize snow storage variations related to orographically-induced rainshadow effects. Compared to the WUS-SR, only SNODAS
(spatial resolution of ~ 1 km) and UA (~ 4 km) reasonably distribute snow storage over windward and leeward watersheds in the Sierra Nevada. Globally-available MR and LR products partition less snow storage on the windward side in both Sierra Nevada and Andes. In the Andes, global products show that more snow water is stored on the leeward side of the mountain than the windward side, completely missing the orographic-rainshadow patterns. Based on these results, to accurately resolve topographically-driven features in snow storage
likely requires spatial resolutions less than ~5 km.

3) Consistent interannual variability is observed among all products assessed in the WUS, whereas there is less agreement in the Andes. This suggests that snow trend studies based on these globally availably snow products applied in the Andes might not be as reliable as those applied in the WUS.

With respect to drivers of uncertainty in snow storage estimates, the key conclusions are:
1) Precipitation primarily explains the variance of snowfall as expected, which propagates to the variance of snow storage. Precipitation uncertainty accounts for a larger portion of snowfall uncertainty in the Andes than the WUS.

2) Aside from precipitation, LSM differences result in varying rain–snow partitioning and snowmelt generation, that play important roles in snow storage variance. Accumulation-season snowmelt generation mechanisms tend





430        to contribute more to the climatological $SWE_{peak}$ uncertainties than the rain–snow partitioning. At coarser spatial

resolutions, there is a larger spatial variance in elevation between products in the Andes that propagates to larger

differences in precipitation falling as rainfall, snowfall loss to ablation, and thus $SWE_{peak}$ than in the WUS where

elevation variance is lower.

435        Data assimilation techniques are used to constrain the SWE uncertainties in SNODAS, UA, WUS-SR and Andes-SR.

Moreover, many products are implicitly constrained by their use of in situ precipitation data in some form over the WUS. With

more accurate precipitation estimates in the WUS, products at HR to MR show reasonable estimates of $SWE_{peak}$. However,

ERA5-Land (MR) and CR products miss orographic-rainshadow patterns (Section 4.1.2). SNODAS and UA generate high

quality SWE estimates in the WUS via inclusion of in situ SWE measurements that are generally unavailable for regions like

the Andes. Additionally, regions like Andes do not have sufficient in situ forcing measurements, resulting in a large uncertainty

in forcings that propagates to SWE.

        Although HR and MR products reasonably estimate snow storage in the WUS, uncertainty in snow storage from

products at coarse spatial resolution in the WUS and at moderate and coarse spatial resolution in the Andes (where there are

limited in situ measurements) need to be reduced to increase their utility for understanding the role of snow in regional water

and energy cycling. Resolving orographic-rainshadow patterns is still a challenging task among existing products. Future work

is needed to reduce the accumulation-season snow storage uncertainty for mountainous regions with limited in situ

measurements. Beyond the accumulation-season processes focused on herein, the snowmelt uncertainty and its drivers in the

melt-season should be investigated to further characterize additional uncertainty in warm-season snowmelt rates and timing.

Aiming for resolving orographic-rainshadow patterns and generating reliable SWE estimates globally will likely require

assimilation of space-borne snow observations. Specifically, future spaceborne missions that can directly provide SWE

measurements at high to moderate spatial resolution over mountainous domains would be extremely valuable for constraining

modeled estimates and generating continuous space–time SWE products at high accuracy over the globe.

## Appendix A. Description of products compared to WUS-SR and Andes-SR

### A1. HR product

455        SNODAS (NOHRSC, 2004) outputs daily SWE from WY 2004 at 1 km over CONUS. Although SNODAS is

available starting from WY 2004, data assimilation was only regularly performed from WY 2005. Therefore, it is suggested

to exclude SNODAS data in WY 2004 for analysis. SNODAS SWE is generated the from NOHRSC Snow Model, an energy

and mass balance model forced with downscaled forcings from numerical weather prediction (NWP) models. Assimilation of

ground-based snow data, airborne SWE from gamma radiation snow surveys, and satellite snow cover is performed via

Newtonian nudging.



## A2. MR products

The UA daily SWE dataset (Zeng et al., 2018; Broxton et al., 2019) at 4 km over CONUS is generated from analysis and interpolation of in situ measurements including SWE from SNOTEL, snow depth, air temperature and precipitation from COOP stations, and gridded estimates including air temperature and precipitation from PRISM. The ordinary kriging method is used for interpolating the ratio of SWE to net snowfall at in situ sites to the PRISM grid. The interpolated ratio is then multiplied by gridded 4-km PRISM net snowfall to get gridded SWE. At in situ sites, precipitation falls as snow on days when snow depth change is positive. As a result, snowfall may be overestimated on rain-on-snow days when both rainfall and snowfall occur, but precipitation is entirely recorded as snowfall. The temperature threshold to partition PRISM precipitation into snowfall and rainfall is interpolated from in situ threshold determined by each site (Broxton et al., 2016a, b). Net snowfall is estimated by the difference in accumulated snowfall and accumulated ablation which is a function of degree days above 0 °C. A new snow density parameterization (Dawson et al., 2017) was developed to convert snow depth at COOP stations to SWE. Precipitation and air temperature for UA are taken from PRISM (Daly et al., 1994).

ERA5 (Hersbach et al., 2020) outputs hourly SWE globally at 0.25° using the H-TESSEL model. An optimal interpolation (OI) method is used to update the grid-averaged snow depth from a maximum of 50 in situ measurements within a radius of 250 km from a given grid cell. In situ snow depth observations from SYNOP and GTS are used as assimilated measurements, and 4-km snow extent from NOAA/NESDIS is applied at elevation lower than 1500 m since 2004. However, SNOTEL/SCAN/COOP snow depth in the WUS are not currently used in the snow assimilation system. Though there might be some sparsely distributed in situ sites that measure snow depth which were assimilated in the WUS and Andes, the impact of data assimilation on SWE in both regions appears to be negligible. The binary snow extent is converted to snow depth at grids below 1500 m, assuming 5 cm of snow depth when snow cover is 1. The conversion is not conducted at elevations above 1500 m to avoid improper terrain information from coarse spatial resolution in mountainous area. SWE is set to 10 m at permanent snow and ice grids. Beyond snow observations, 4-km precipitation data from NCEP stage IV over the U.S. was assimilated in ERA5 using 4D var data assimilation method (Lopez, 2011). NCEP precipitation data is produced radar and gauge observations (Lin and Mitchell, 2005). Hence it is reasonable to assume that ERA5 precipitation may be more accurate over the WUS than the Andes where such data is not assimilated.

## A3. CR products

ERA5-Land (Muñoz-Sabater et al., 2021) hourly SWE globally at 0.1° is generated from the same land surface model as ERA5 (with different versions) but driven by downscaled and lapse rate corrected forcings from ERA5 at higher spatial resolution. Specifically, shortwave, longwave, liquid, and solid precipitation are downscaled using a linear triangular mesh interpolation. Other variables such as air temperature, specific humidity, relative humidity, and surface pressure are adjusted to account for differences in elevations between the two spatial resolutions. No additional data assimilation is involved in generating the ERA5-Land SWE.



The suite of GLDAS 2.1 products consist of daily SWE since WY 2001 from four globally distributed products (Rodell et al., 2004). The four products are generated from three LSMs and at two spatial resolutions (i.e., Noah LSM at 0.25°:

GLDAS – NOAH025; Noah LSM at 1.0°: GLDAS – NOAH10; VIC LSM at 1.0°; GLDAS –VIC10; Catchment LSM at 1.0°: GLDAS – CLSM10). The same meteorological forcings from multiple sources, including NOAA/GDAS, GPCP1.3, and corrected AGRMET, are employed to generate the four products. Adjustments for forcings are conducted to account for the elevation differences between GLDAS at 1.0° and 0.25°. No snow data assimilation is conducted in generating the products, whereas input forcings include sources from in situ measurements.

MERRA2 outputs hourly SWE globally at 0.625° x 0.5° resolution using the Catchment LSM (Reichle et al., 2017). The Catchment LSM is forced by bias-corrected precipitation using Climate Prediction Center (CPC) unified gauge-based analysis of global daily precipitation products. Similar to the GLDAS subset, no snow data assimilation is involved in generating the MERRA2 dataset, whereas in situ precipitation measurements are involved in deriving the SWE.

**Data availability**

The Andes-SR and WUS-SR datasets are publicly available on https://ucla.app.box.com/folder/89588138091?v=ANDES-SWE-REANALYSIS (last access: 16 November 2022) and https://doi.org/10.5067/PP7T2GBI52I2 (last access: 6 September 2022), respectively. SNODAS product is available on https://doi.org/10.7265/N5TB14TC. The UA daily SWE product is available on https://doi.org/10.5067/0GGPB220EX6A (last access: 6 December 2022). The PRISM product is available on https://ftp.prism.oregonstate.edu/ (last access: 17 October 2022). The ERA5 product is available on

https://doi.org/10.24381/cds.adbb2d47 (last access: 24 September 2022). The ERA5-land product is available on https://doi.org/10.24381/cds.e2161bac (last access: 25 October 2022), respectively. For GLDAS 2.1 products, the GLDAS – NOAH025 is available on https://doi.org/10.5067/E7TYRXPJKWOQ (last access: 17 October 2022); GLDAS – NOAH10 is available on https://doi.org/10.5067/IIG8FHR17DA9 (last access: 17 October 2022); GLDAS – VIC10 is available on https://doi.org/10.5067/ZOG6BCSE26HV (last access: 17 October 2022); and GLDAS – CLSM10 products are available on

https://doi.org/10.5067/VCO8OCV72XO0 (last access: 17 October 2022). MERRA2 SWE is available on https://doi.org/10.5067/RKPHT8KC1Y1T (last access: 16 August 2022); bias-corrected precipitation is available on https://doi.org/10.5067/7MCPBJ41Y0K6 (last access: 19 December 2022); bias-corrected snowfall is available on https://doi.org/10.5067/L0T5GEG1NYFA (last access: 19 December 2022).

**Author contribution**

YF contributed to data acquisition and analysis, manuscript conceptualization and writing. YL contributed to data acquisition, manuscript conceptualization and revision. DL contributed to manuscript conceptualization and revision. HS contributed to manuscript conceptualization and revision. SM contributed to manuscript conceptualization, revision, and supervision.



**Competing interests**

The authors declare no competing interests.

**Acknowledgements**

This work is funded by the National Science Foundation (NSF) Grant # 1641960, the National Oceanic and Atmospheric Administration (NOAA) OAR/OWAQ Observations Program Award # NA18OAR4590396, and the National Aeronautics and Space Administration (NASA) IDS Grant # 80NSSC20K1293.

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
