# Peer review of "Spatiotemporal snow water storage uncertainty in the midlatitude American Cordillera"

_EGUsphere, 2023_

## Author Response (AR1)

We thank the editor and both reviewers for their thorough reviews and valuable comments on the manuscript. The responses to reviewer comments (black text) are shown in blue font, added and revised manuscript text is  shown in red font, and any original manuscript text is shown in gray font. Figure R1 – R4 are used only in the response document.

**Reply on Editor:**

Data are available but code is not, making it difficult to fully assess the methods used in the reanalysis. For example, I still don't understand how clouds are masked. It's difficult to imagine there haven't been code updates in the fsca retrievals over the past 9 years since the Cortés et al. (2014). Closed source code goes against TCD recommendations.

Authors are encouraged to deposit software, algorithms, and model code in FAIR-aligned repositories/archives whenever possible. These research outputs are then cited in the manuscript using the received DOI and included in the reference list. The manuscript must then include a section entitled "Code availability" or, in the case of data and code, "Code and data availability".

https://www.the-cryosphere.net/submission.html

We appreciate the comment regarding data and software. The methodology to generate the WUS and Andes snow reanalysis datasets are documented in Margulis et al. (2016, 2019), and the previously published datasets themselves are presented in Fang et al. (2022) and Cortes and Margulis et al. (2017) respectively, where all required data/software protocols were followed. Given that this paper is an intercomparison of previously published datasets, we do not think providing the reanalysis code is necessary in this context, since no new datasets were generated; only existing datasets were post-processed for this work. Moreover, the reanalysis code used previously to generate these datasets was designed specifically to run on UCLA high-performance cluster and is likely therefore not very practical for distribution at this point in time. However, in principle we agree with the comment and are working on a more general and practical version of the reanalysis code for future distribution with the next major release/s of the reanalysis datasets.

Regarding the cloud masks, the method to mask out clouds in the development of the WUS snow reanalysis is described in Fang et al. (2022):

"Following the cloud screening methods described in Margulis et al. (2019) and Liu et al. (2021) the internal Landsat cloud mask is used to attempt to exclude images with cloud cover fraction greater than 40%. For those images included, the internal cloud masks are used to screen out any cloudy pixels."

So we use the built-in Landsat cloud mask at the tile level as the first screening (to either include/exclude a particular Landsat tile) and then for those included tiles we use the built-in mask to include/exclude specific pixels.

Resolutions and differences between the Andes SR and WUS SR (180 m vs 480 m) need more explanation. According to the NSIDC link for the WUS SR dataset (https://doi.org/10.5067/PP7T2GBI52I2), the resolution is 16 arc-sec/0.004 deg (~500 m).

The Andes SWE reanalysis is accessible, but should be in a repository with a DOI. The readme for that one says it's 180 m for the raw data, but re-gridded to a lat/lon grid at 0.001 deg (~100 m). Table 1 says 16"/500 m for WUS-SR and 6"/180 m for ANDES-SR. Please clean up these inconsistencies.

We apologize for the discrepancies and have fixed them as indicated below. One other key difference is that the Andes-SR was only applied to seasonal snow pixels (i.e., glacier screening was done prior to applying the reanalysis). In the more recently derived WUS-SR, all pixels are run so that any "non-seasonal" pixels can be screened out after the fact. The other key point related to the comment is that for both the Andes-SR and the WUS-SR, only SWE outputs are available.

The spatial resolution of "~500 m" was used in the NSIDC link as an order of magnitude reference, however, the resolution of WUS-SR is closer to 480 m.

Based on your suggestion we are storing the Andes-SR to another repository with a DOI https://doi.org/10.5061/dryad.ngf1vhj0s, which is in process and has the temporary URL https://datadryad.org/stash/share/jk_o2Y1vXyB4Yj0tXLdMWMlyB9zVDMNVcvmGsIwJ7PM. Note that the DOI will be activated, and the temporary URL will be deactivated once the dataset passes the review process. If necessary, we can ensure this is completed prior to acceptance/publication.

We summarized the key differences relevant to this paper and have revised and added the following sentence in the main text line 90-94 for clarification:
"The Andes-SR (WYs 1985 to 2015; Cortés and Margulis, 2017) SWE estimates were derived on a regular 180 m resolution grid before regridding to a regular latitude/longitude grid (0.001 ° or ~100 m). The WUS-SR (WYs 1985 to 2021; Fang et al., 2022) SWE estimates were derived on a regular lat/lon grid (16 arcseconds or ~ 480 m resolution). The different resolutions used for the Andes and WUS domains were based on computational constraints. In addition to spatial resolution differences, glaciers and elevation below 1500 m were masked out before applying the Andes-SR. The newer WUS-SR dataset is applied over the full domain and then masked afterwards as described in Fang et al. (2022)."

I still suggest including areas with persistent snow/ice, especially the melt, but I've brought that up in previous reviews.

Thank you for the suggestion. As mentioned above, the Andes-SR was generated for seasonal snow areas only, i.e., pixels deemed as glaciers were not included. For the WUS-SR, only SWE is output and therefore we cannot provide snowmelt estimates. So the available reanalysis data is 1) Andes-SR *SWE* in seasonal snow pixels and 2) WUS-SR *SWE* in all pixels (masked afterward).

We agree that persistent snow/ice and melt are important topics for snow hydrology, but this should be done in future work using a more consistent approach across products. Future versions of the snow reanalysis datasets will include snowmelt and forcings.

Section 4.3 & Fig. 9-11 - Section says "all products" included but Figs are missing Andes/WUS SR. Why?

Thank you for the comment. In the manuscript we differentiate the reanalysis ("reference") datasets from the other datasets ("products") in line 114-116:

"Globally and regionally available datasets are referred to as "products" to distinguish them from the reference "datasets", i.e., WUS-SR and Andes-SR."

Section 4.3 and Figure 9 to 11 involves forcings including precipitation and snowfall. These variables were not output in the Andes-SR and WUS-SR datasets (only SWE is available). Hence for results in Section 4.3, only reference SWE are used as benchmarks.

**Reply on Review #1**

MAJOR COMMENTS

1. The paper's goal is to characterize uncertainty in snow water storage. This is more feasible when done across all scales (10 products in WUS, 8 products in Andes), however I think there are too few datasets to assess uncertainty within a given spatial resolution (e.g., HR and MR). In other words, having only 1 or 2 models in a given resolution does not make it possible to quantify uncertainty with confidence (i.e., it becomes one model versus another). This is not the fault of the authors per se, as they are generally using what is readily available. While recognizing the significant work that has already been done, I might suggest adding additional HR and MR datasets as feasible. For WUS, two readily available and well-known daily SWE datasets are SWE reconstruction at 500 m (e.g., Bair et al., 2023) and DayMet SWE at 1 km (e.g., Thornton et al., 2022). Including other SWE datasets such as these would help to better characterize uncertainty at finer spatial scales and would represent more distinct approaches that are not currently represented in the study (e.g., SWE reconstruction). I would argue that having a more comprehensive sampling of existing SWE datasets would elevate the utility of the paper to the community.

We appreciate this comment. Our categorization of datasets as HR, MR, and LR were not designed to make definitive conclusions based on resolution (for reasons pointed out by the reviewer), but rather as a framework for discussion and to point at some general variations that seem to occur as a function of resolution. Given that the Bair et al. (2023) dataset came out after submission of this paper, we would argue that the extra analysis requested is a heavy burden that could instead be done in future work. As for the DayMet product, the modeled SWE is based on a simple temperature driven model and therefore is in a bit of a different class than the other products (and would also require a significant amount of more analysis for the paper). We agree that further analysis could be warranted and therefore reference both datasets in the conclusions as datasets that should be examined in the context of this and other intercomparison analysis (along with other datasets that will continue to become available).

"New and future SWE products such as the recently published SWE reconstruction at 500 m (Bair et al., 2023), and other products such as DatMet SWE at 1 km (Thornton et al., 2022) could be examined to further characterize uncertainty in higher resolution products."

Instead of adding new datasets we have performed the additional analysis in response to suggestions shown below.

2. The SWE product intercomparison focuses on the "snow accumulation season" (L. 135-145), which is defined as the period before peak SWE (L. 139-140). However the accumulation season is not always well-defined in all years, locations, and spatial scales. For instance, snow may be more intermittent in lower elevations, in drier years, and/or at coarser spatial scales. Notably, the timing of peak SWE varies in these cases (as across the products in Figure 2), which suggests that the uncertainty in snow water storage may be larger at other times in the year (e.g., March 1 in WUS). Hence, I am wondering about whether peak SWE is necessarily the most robust way to characterize uncertainty across snow products? In addition to the analyses presented, it could be helpful to characterize the uncertainty in time (e.g., by dowy) rather than just by a fixed point (e.g., peak SWE).

Thank you for your comment. The rationale for focusing on the pixel-wise accumulation season is that much of the uncertainty comes from the accumulation season and that accumulation errors/differences then propagate to the melt season. It is a valid point that the accumulation season definition may not be optimal for intermittent snow and/or snow in extremely dry years. However, we use the same definition of accumulation season for all years, locations, and spatial scales to maintain consistency over multiple products. It is expected that the key results do not qualitatively change if another proxy date for peak SWE is used (shown below). Hence, we would prefer to keep the current focus on (pixel-wise) peak SWE and the accumulation season in the main manuscript text.

Additional analyses (added in the SI) for March 1st and April 1st SWE confirm that similar uncertainty conclusions are reached as those when using pixel-wise peak SWE.

"S4. Climatological March 1st SWE and April 1st SWE

Overall, the relative uncertainties of climatological $SWE_{peak}$ over the WUS (Figure 3k) and Andes (Figure 4i) are consistent with uncertainties of March 1st and April 1st SWE across different products (Figure S6). In the WUS, HR and MR products generally agree with the WUS-SR, whereas LR products underestimate SWE. The WUS-SR average $SWE_{peak}$, March 1st and April 1st SWE values are 269, 185 and 150 km$^3$, respectively. In comparison, the average $SWE_{peak}$, March 1st and April 1st SWE values from HR and MR products are 284, 185 and 168 km$^3$, respectively. Thus, for HR and MR products, March 1st SWE has the lowest bias (0%) followed by $SWE_{peak}$ (overestimated by 6%), and April 1st SWE (12%). For LR products, the average $SWE_{peak}$, March 1st and April 1st SWE values are 127, 75 and 43 km$^3$, respectively. The lowest bias is from $SWE_{peak}$ (underestimated by 53%), followed by March 1st (59%) and April 1st SWE (71%).

In the Andes, Andes-SR shows that $SWE_{peak}$ is 29 km$^3$, March 1st SWE is 26 km$^3$ and April 1st SWE is 24 km$^3$. The average values for MR and LR are 19, 14 and 13 km$^3$, respectively. $SWE_{peak}$ has the lowest bias (34%), followed by March 1st and April 1st SWE with the same level of bias (46%).

[Figure]

Figure S3. Climatological March 1st SWE (top panels) and April 1st SWE (bottom panels) over the WUS (left panels) and Andes (right panels). Black error bars represent the interannual inter-quartile range (IQR)."

3. Section 4.1 analyses spatial variations in peak SWE across the study regions and with respect to windward/leeward basins. One aspect that would be useful to analyze and compare across products is the lapse rate in peak SWE across the windward/leeward sides. While peak SWE is lower in the LR products and higher in the HR products (Figures 5-6), it must be remembered that the LR products have less variation in elevation than the HR products. As such, I think this should be normalized in order to assess how the elevation gradients in peak SWE compare across products. This would be potentially important to know for certain applications (e.g., downscaling a LR product to higher resolution).

Thank you for your suggestion regarding the lapse rate in peak SWE. Additional analysis and Figure 7 have been added in the main text as shown below:

"The elevational distributions of bin-averaged climatological $swe_{peak}$ in the Sierra Nevada (Figure 7a) and Andes (Figure 7b) are plotted to compare the elevational gradient of windward and leeward $swe_{peak}$ from products with different spatial resolutions. The lapse rate in $swe_{peak}$ was determined by linear regression of $swe_{peak}$ averaged across elevational bins (Text S5). Lapse rates from GLDAS products at 1.0° are not included because the subdomains analyzed are covered by less than 10 pixels (Figure S7 and S8).

Based on the WUS-SR, climatological $swe_{peak}$ on the windward side of the Sierra Nevada monotonically increases up to ~3.5 km. Across different products, the uncertainty of $swe_{peak}$ is smaller at the lower elevation ~ 1-1.5 km, however, the differences in lapse rate project to larger $swe_{peak}$ uncertainty as elevation increases. The gradients of windward $swe_{peak}$ (i.e., $d(swe_{peak})/dz$) from WUS-SR, averaged over HR and MR products, and averaged over LR products are 0.40 m/km, 0.38 m/km, and 0.10 m/km, respectively. On the leeward side of the Sierra Nevada, the $swe_{peak}$ increases monotonically with elevation from 1 – 3.5 km in the WUS-SR and most of the other products. Similarly, the uncertainty of $swe_{peak}$ is smaller at low elevation from 1 – 1.5 km and gradually increases with elevation corresponding with the differences in lapse rate across different products. The gradients of leeward $swe_{peak}$ (i.e., $d(swe_{peak})/dz$) from WUS-SR, averaged over HR and MR products, and averaged over LR products are 0.22 m/km, 0.23 m/km, and 0.13 m/km, respectively. HR and MR products have qualitatively similar elevational distributions of $swe_{peak}$ on both the leeward and windward side of the Sierra Nevada for elevations below 3 km, whereas that $swe_{peak}$ from LR are underestimated with large differences in lapse rates compared to WUS-SR.

[Figure]

Figure 7. Elevational distribution of windward and leeward swe$_{peak}$ in the Sierra Nevada and Andes across reference datasets and products with spatial resolution higher than 1°. Each dot represents the elevation bin-averaged swe$_{peak}$. The interval of each bin is set to be 0.5 km. GLDAS products at 1° are not included for comparison due to too few points. On the windward side of the subdomains, dots within the red shaded areas are used to compute lapse rates. On the leeward side, dots in the darker shaded areas are used to compute lapse rates.

On the windward side of the Andes, swe$_{peak}$ from the Andes-SR increases from 1.5 – 3 km, with decreases between 3 and 5 km. The swe$_{peak}$ uncertainty is smaller at low elevation bands between 1.5 - 2 km. The uncertainty gets larger as elevation increases from 2 – 3 km corresponding to large differences in positive lapse rates. In contrast, large differences in negative lapse rates above 3 km reduces the uncertainty as elevation increases. The lapse rates of windward swe$_{peak}$ from the Andes-SR are 0.4 m/km between elevation bands of 1.5 – 3 km and -0.16 m/km between 2.5 – 5 km (Table S1). On the leeward side, swe$_{peak}$ increases between 1.5 – 4 km and slightly decrease above 4 km in the Andes-SR. Similar to the windward side, differences in positive lapse rate below 3 km project to larger swe$_{peak}$ uncertainty as elevation increases from 1.5 km, whereas differences in negative lapse reduces uncertainty as elevation increases above 3 km. The lapse rates of windward swe$_{peak}$ from the Andes-SR are 0.27 m/km between elevations of 1.5 – 3 km, and -0.03 m/km between 3.5 – 5 km."

In the SI, we added a table of lapse rates and plots of the elevational distribution swe$_{peak}$ for all products.

"Text S1. The lapse rates were determined based on linear regressions across elevational bins in the Sierra Nevada and Andes based on the swe$_{peak}$ distribution from the snow reanalysis datasets (Table S1). Specifically, swe$_{peak}$ increases with elevation on both the windward and leeward side of the Sierra Nevada

from 1 – 3 km. In the Andes, swe$_{peak}$ increases with elevation over 1.5 – 3 km on both sides of the Andes, whereas it decreases with elevation over 2.5 – 5 km on the windward side and 3.5 – 5 km on the leeward side. Figure S7 and S8 shows that GLDAS products at 1° do not have enough data points to compute the lapse rates and therefore are excluded in the analysis.

Table S1. Derived swe$_{peak}$ lapse rates over the Sierra Nevada and Andes across different elevational bands. The unit of lapse rate is m (SWE)/km (elevation) with a positive value representing an increase of swe$_{peak}$ in meters per increase of elevation in kilometers.

| Domain | Sierra Nevada | | Andes | | | |
|---|---|---|---|---|---|---|
| | Windward | Leeward | Windward | Leeward | Windward | Leeward |
| Elevation | 1 – 3 km | | 1. 5 – 3 km | | 2.5 – 5 km | 3.5 – 5 km |
| WUS-SR/ Andes-SR | 0.40 | 0.22 | 0.40 | 0.27 | -0.16 | -0.03 |
| SNODAS | 0.45 | 0.26 | - | - | - | - |
| UA | 0.43 | 0.26 | - | - | - | - |
| ERA5-Land | 0.25 | 0.18 | 0.34 | 0.44 | -0.21 | -0.17 |
| ERA5 | 0.14 | 0.16 | 0.07 | 0.26 | -0.08 | -0.18 |
| GLDAS-NOAH025 | 0.05 | 0.08 | 0.09 | 0.13 | -0.02 | 0.06 |
| MERRA2 | 0.13 | 0.15 | 0.03 | 0.03 | -0.03 | 0.02 |

[Figure]

Figure S7. Elevational distribution of windward and leeward swe_peak across the Sierra Nevada. The black dots are bin-averaged swe_peak values.

[Figure]

Figure S8. Elevational distribution of windward and leeward swe$_{peak}$ across the Andes. The black dots are bin-averaged swe$_{peak}$ values."

GENERAL COMMENTS

- Please make consistent use of the acronym for the low/coarse resolution products. Sometimes it is "CR" and sometimes "LR". Please select one convention only and use it consistently.

Thanks for catching this error. We replaced "CR" with "LR" throughout the manuscript.

LINE COMMENTS

- Line 57: It may be worth noting the mountain ranges are also disparate with respect to elevation.

This sentence has been revised to:
"The WUS and Andes domains have comparable atmospheric circulation patterns and hydrologic cycles (Rhoades et al., 2022), but are disparate with respect to elevation and the amount of available in situ information."

- Line 75: Delete "shows that".

"shows that" has been deleted.

- Line 112: Add "satellite snow cover" before "observations".

"satellite snow cover" has been added before "observations":

"The snow reanalysis reference datasets are, by design, constrained by satellite snow cover observations using a data assimilation approach."

- Line 131-134: I think this climatological analyses could be of interest, and would request their inclusion in the supplement document.

We added climatological analysis from WYs 2004 to 2021 in WUS and from WYs 2001 to 2015 in the Andes in the supplemental information. The additional analysis is shown below:

"In the WUS, GLDAS products are only available over Water Years (WYs) 2001 to 2021, and SNODAS is only available over WYs 2004 to 2021, while all other products span the 37-year record (WYs 1985 to 2021). In the Andes, the GLDAS products are only available over WYs 2001 to 2015, while all other products span the 31-year record (WYs 1985 to 2015).

The climatological SWE over the longer study periods agrees well with climatological SWE over the shorter periods (Figure S1). In WUS, climatological SWE from SNODAS, UA and ERA5-Land are comparable with WUS-SR for either time period used whereas other products underestimate SWE volumes in both cases. In the Andes, over both time periods, ERA5-Land overestimates SWE, ERA5 generates comparable SWE, and the other products underestimates SWE compared to the Andes-SR.

[Figure]

Figure S1. Climatology of seasonal cycle of SWE volume in the WUS over WYs 1985 – 2021 (a) and WYs 2001 – 2021 (b), and Andes over WYs (c) and WYs 2001 – 2015 (d). Solid lines represent high-resolution (HR) datasets and products, dashed lines represent moderate-resolution (MR) products, and dotted lines represent low-resolution (LR) products."

- Line 171: Delete "choose to".

"choose to" has been deleted.

- Line 179: I would think that all three resolutions (HR plus MR and LR) may straddle both windward and leeward watersheds rather than just the MR and LR resolutions. I assume you would see this if you zoomed in more in Figures S3a and S4a. Also, replace "CR" with "LR" here?

Thank you for your comments. It is true that all three resolutions would straddle both windward and leeward watersheds to varying degrees. For MR and LR products, fractional $swe_{peak}$ is aggregated to get $SWE_{peak}$ for each watershed. For HR products, we simply aggregate the full $swe_{peak}$ that is within the watershed shapefile and did not compute the fractional $swe_{peak}$. For HR products, the spanning of windward/leeward sides has a negligible impact on the overall distribution. We revised the sentence as below to avoid confusion:

"Since  pixels may cover both windward and leeward watersheds, for MR and LR products, fractional $swe_{peak}$ is aggregated to get $SWE_{peak}$ over the two types of watersheds separately (Fig. S4 and S5). For HR products and datasets the pixels spanning the windward to leeward side has a negligible impact on the distribution."

- Line 305 and 330: I find the titles for sections 4.3.1 and 4.3.2 to be odd. Consider reducing and rewording.

Thank you for your suggestion. We have modified them for clarity as:

"Impact of accumulation-season precipitation and snowfall on annual $SWE_{peak}$

Impact of LSM and spatial resolution on climatological $SWE_{peak}$"

- Line 310: Replace "more" with "higher".

"more" has replaced with "higher".

- Line 326: Replace "is" with "are".

"is" has been replaced with "are".

- Lines 359-364: I would suggest elaborating a little more here on model differences.

Thank you for your suggestion, we have elaborated the text:

"The differences in $R_{acc}/P_{acc}$ are $\leq 0.1$ between GLDAS-VIC10 and GLDAS-CLSM10 in contrast to the differences of 0.2-0.3 in $A_{acc}/S_{acc}$. GLDAS-VIC10 tends to have higher $P_{acc}$, $S_{acc}$, and $SWE_{peak}$ which are closer to those from the HR or MR snow products. The better performance of GLDAS-VIC10 than others might be associated with the usage of snow elevational bands in the VIC model, in which sub-grid snowfall and SWE estimates are better represented. GLDAS-CLSM10 has the highest rates of $A_{acc}/S_{acc}$ and lowest $SWE_{peak}$. Previous study shows a larger portion of snowfall is lost as accumulation-season ablation in the Catchment model (Xiao et al., 2021). Therefore, a better characterization of snowmelt during the accumulation season is beneficial to improve $SWE_{peak}$ accuracy."

- Line 400: Replace "less" with "fewer".

"less" has been replaced with "fewer"

- Lines 448-452: I think these sentences are not well justified and need to either be removed or better connected to the study. The study does not suggest why future/new spaceborne data are needed to assess SWE in these mountain ranges. This conclusion might have been better motivated if an existing spaceborne sensor that maps SWE (e.g., passive microwave) had been included. Multiple SWE datasets in this paper utilize existing spaceborne snow cover data (e.g. reference and SNODAS) and appear to capture certain spatial patterns like the rain shadow effect.

Thank you for your suggestions. We rephrased these sentences to suggest the potential to use spaceborne snow measurements to constrain model-based snow estimates.

"The ability to capture orographic rainshadow patterns from snow reanalysis datasets and SNODAS encourages the usage of existing spaceborne snow covered area measurements and/or future spaceborne missions that can directly provide high-resolution SWE measurements to constrain mountain SWE."

- Lines 473-492: It appears the ERA5 paragraph (Lines 473-485) needs to be swapped with the ERA5-Land paragraph (Lines 487-492) based on their resolutions (ERA5-Land is a MR product, ERA5 is a LR product).

Thank you for catching this. We swapped ERA-Land and ERA5. ERA5 has been placed in the LR section and ERA5-Land is in MR section.

TABLE AND FIGURE COMMENTS

- Figure 1: Suggest labeling the Cascades in the WUS map since the text references them in multiple places.

Thank you for the suggestion. We have labeled the Cascades in Figure 1.

- Figure 7b-c: There appears to be an interesting outlier year where UA and ERA5-Land have much lower peak SWE than WUS-SR. This appears to be a high snow accumulation year. Can you please identify which year this is in the text and provide a brief discussion point about it? These products have greater correspondence to WUS-SR in most other years, so this year may be negatively skewing the error statistics.

The outlier is WY 1993 which is identified as the wettest year in WUS-SR record. It is not clear to us why UA and ERA5-Land disagree with the WUS-SR in this year. A comparison of WUS-SR and in situ peak SWE in WY 1993 shows that WUS-SR agrees with the in situ SWE with a correlation coefficient of 0.76. A negative mean difference suggests that $SWE_{peak}$ from WUS-SR is slightly lower than that from in situ data. The reanalysis performance for this year is comparable to other years and the overall verification results.

[Figure]

Figure R1. scatter plot of peak SWE from Reanalysis SWE and in situ SWE from SNOTEL and CDEC over the WUS in WY 1993.

Figure R2 shows that the statistics do not change much by removing the WY 1993 data points. The R values are slightly improved from 0.90 in UA and 0.91 in ERA5-Land to 0.92, whereas MD and RMSD are larger compared to the original plot including WY 1993. We added the description of the outlier year in the Figure Caption:

"Figure 8. Scatter plots (a – i) of SWE$_{peak}$ volumes between WUS-SR and other products. Each dot represents SWE$_{peak}$ volume (km$^3$) for each year over the study period (WYs 1985 to 2021) where data are available. For the SNODAS and GLDAS products, the comparison is over WYs 2005 to 2021, and 2001 to 2021, respectively. The WY 1993 SWE$_{peak}$ in WUS-SR is the highest and much higher than those from UA and ERA5-Land. Statistics do not change significantly if excluding this data point. (j) shows the SWE$_{peak}$ percentiles in each WY over the overlapping period including all products (WYs 2005 to 2021)."

[Figure]

Figure R2 Same as Figure 7 but removing WY 1993.

- Figure 7 caption (line 296): Replace "is" with "are".

"is" has been replaced with "are".

- Figures 7j and -8a: It seems for the heat maps, a calculation of the spearman rank correlation would be useful to assess the agreement in dry to wet years for each product vs. the reference.

The spearman rank correlation for the SWE$_{peak}$ percentiles have been listed in Table 2 with descriptions shown below:

"Overall, dry to wet years identified from products in the WUS generally agree with the WUS-SR with a correlation coefficient above 0.8 over WYs 2005 to 2021 (Table 2). In contrast, discrepancies are evident among SWE$_{peak}$ percentiles computed from different products over WYs 2001 to 2021. Percentiles from ERA5-Land and GLDAS-NOAH025 agree well with the Andes-SR. However, the correlation is low between other products and Andes-SR. Although SWE$_{peak}$ from ERA5 has comparable climatology with Andes-SR (Figure 4i), its interannual distribution disagrees with the Andes-SR, especially after WY 2001."

Table 2. Correlation of SWE$_{peak}$ percentiles of each product against the reference datasets over WYs 2005 to 2021 in the WUS, and WYs 2001 to 2021 in the Andes.

| Products | WUS-SR | ANDES-SR |
|---|---|---|
| SNODAS | 0.89 | - |
| UA | 0.86 | - |
| ERA5-Land | 0.91 | 0.93 |
| ERA5 | 0.95 | 0.11 |
| GLDAS-NOAH025 | 0.92 | 0.85 |
| MERRA2 | 0.87 | 0.51 |
| GLDAS-VIC10 | 0.95 | 0.60 |
| GLDAS-NOAH10 | 0.91 | 0.42 |
| GLDAS-CLSM10 | 0.84 | 0.46 |

- Figure 10: It would be helpful to include a dashed line for the t_peak (DOWY) of the reference data.

We have added the t_peak (DOWY) values of WUS-SR and Andes_SR as solid red lines in Figure 10 as shown below.

[Figure]

Figure 10. Climatological $SWE_{peak}$, $S_{acc}$, and $P_{acc}$ volumes aggregated over WUS (top panel) and Andes (bottom panel) in $km^3$. Red triangles (corresponding to right y-axis) show the $t_{peak}$ averaged over all pixels and WYs. The horizontal dashed lines and red lines are the reference snow reanalysis SWE volumes and $t_{peak}$, respectively, from WUS-SR and Andes-SR. The vertical dashed lines group the products by spatial resolution (i.e., HR, MR, LR). The black text lists the $A_{acc}/S_{acc}$ and gray text lists the $R_{acc}/P_{acc}$.

**Reply on Review #2**

General Comments:

1. It is difficult to appreciate the water storage units of cubic kilometers and to put the climatological peak and uncertainty metrics in the context of water resources. It seems that all reservoirs in the contiguous US hold 600 km3 of water (Steyaert et al., 2022). This suggests that the climatological average snow water storage in the western US is 269/600 or 45% of all reservoir storage in the contiguous US (much of which is in the western US). While this is a compelling number, the more compelling result, in my opinion, would be expressing the uncertainty of global models relative to this US reservoir storage estimate. My quick assessment (check this) is that the low-resolution products underestimate snow volume by nearly 24% of all the water held in these US reservoirs. That astounding fact is likely not appreciated by most users of those (commonly used) data.

Thank you for the great suggestions and providing the sources. We verified the number and percentage you computed are correct. We included this information in the conclusion section:

"In the WUS, HR and MR snow products are in better agreement with WUS-SR peak snow storage (269 km$^3$) than the LR snow products, where the snow storage is biased low with large uncertainty. The climatological snow storage was found to be 284 km$^3$ ± 14 km$^3$ among HR and MR products and 127 km$^3$ ± 54 km$^3$ among LR products. For context, the reservoir capacity in the contiguous U.S. is around 600 km$^3$ (Steyaert et al., 2022). Thus, based on the WUS-SR, the snow water stored in the WUS is 45 % (269 km$^3$ of WUS-SR SWE$_{peak}$ / 600 km$^3$ of contiguous US reservoir capacity) of the total reservoir capacity. Compared to the snow storage from WUS-SR, the averaged snow water storage from LR products misses 142 km$^3$ of snow water storage, equivalent to 24% of total reservoir capacity over the contiguous U.S."

Steyaert, J.C., Condon, L.E., WD Turner, S. and Voisin, N., 2022. ResOpsUS, a dataset of historical reservoir operations in the contiguous United States. Scientific Data, 9(1), p.34.

2. Please discuss the implications of snow model uncertainty in coarse scale model (> 10 km) applications on the topic of snow volume sensitivity to warming. For example, Siirla-Woodburn et al. (2021) sited in this paper uses coarse-scale model output and concludes a dire water resource scenario for mid-century. Might results of such studies be different and arguably more accurate if models were run at finer spatial resolution?

Thank you for suggesting the analysis of snow volume sensitivity to warming in coarse scale models. We agree that, given the underestimated SWE, it would disappear more quickly from coarse resolution models if melt rates were the same. The following comments are added in the conclusion:

"The averaged SWE volumes from LR products in the WUS and Andes are underestimated by over 30% compared to the reanalysis datasets. For similar melt rates, SWE computed from LR models would therefore disappear more quickly than HR/MR products. Hence calculation of snow volume sensitivity based on LR products could exaggerate the impact of warming on snow loss."

Additionally, we computed the snow volume loss trends. However, we found that WUS-aggregated snow trends are not significant (p-value > 0.05) over all the products and that the snow loss rates vary significantly (Figure R3), likely in part due to the relatively short analysis periods.

Moreover, the slope and p values of the fitted trendlines are sensitive to the study period chosen. For example, if the starting year is a wet year (Figure R4), the p-value and slope (△) would be much lower

than starting with a normal or dry year (Figure R3). Therefore, we believe this topic deserves further investigation, but that it is beyond the scope of this paper.

[Figure]

Figure R3. WUS-aggregated peak SWE trend. △ represents the snow loss rate computed using Theil–Sen slope. P-value is computed based on Mann Kendall test. The study periods for GLDAS, SNODAS and the rest of products and dataset are WY 2001 to 2021, 2005 to 2021, and 1985 to 2021 respectively.

[Figure]

Figure R4. WUS-aggregated peak SWE trend. △ represents the snow loss rate computed using Theil–Sen slope. P-value is computed based on Mann Kendall test. The study periods for GLDAS, SNODAS and the rest of products and dataset are WY 2001 to 2021, 2005 to 2021, and 1994 to 2021 respectively.

Siirila-Woodburn, E. R., Rhoades, A. M., Hatchett, B. J., Huning, L. S., Szinai, J., Tague, C., Nico, P. S., Feldman, D. R., Jones, A. D., Collins, W. D., and Kaatz, L.: A low-to-no snow future and its impacts on water resources in the western United States, Nat Rev Earth Environ, 2, 800– 819, https://doi.org/10.1038/s43017-021-00219-y, 2021.

Detailed Edits:

Line 200: To make the comparison clear, perhaps add "in the Andes than they do in the WUS".

"in the Andes" has been added before "than they do in the WUS".

---

## Author Response (AR2)

We would like to thank both editor and reviewer for their comments on the revised manuscript. The responses to reviewer comments are shown in blue font, proposed additions and revisions of the manuscript are shown in red font, and any original manuscript text is shown in gray font.

**Editor**

Dear Dr. Fang:

Please address the comments by Referee #1 on Figure 7 of the revised manuscript. Also change 1.11022e-16 to 0 on x-axis in the lower sub plot.

Response:

Thank you for your comments. We have addressed the comments by Referee #1 on Figure 7 and changed the 1.11022e-16 to 0 on x-axis in the Andes panel on Figure 7.

**Referee #1**

SUMMARY AND OVERALL RECOMMENDATION

The authors were receptive to the critiques I provided and have delivered reasonable responses to the comments/concerns I raised in my initial review. In particular, they have provided additional supplementary analyses justifying certain aspects of their work (e.g. peak SWE vs. April 1 and March 1 SWE) and more detailed analyses on SWE-elevation relationships in the main text (Section 4.1.2, Fig. 7). As I indicated previously, I think the community will find value in this paper, and it should be published once all outstanding comments are resolved. I offer two final minor corrections/comments.

Minor/technical corrections:

- Figure 7 caption: Please include "a" and "b" in the figure panels and reference "a" and "b" in the caption (as is done in the main text).

Response: Thank you for your suggestion. We added "a" and "b" in the figure panels and in the caption.

- Figure 7 and L. 304-312: The shape of the windward SWE distribution with elevation for Andes-SR is quite odd to me, particularly above 4.5 km elevation, and I think warrants some commentary (currently the discussion focuses only on the lower elevations in the Andes). I can understand SWE increasing with elevation until a certain elevation and then decreasing at higher elevations due to limitations in atmospheric moisture availability (as appears to be the case in the WUS). However, what can possibly explain the increase in SWE from 4.5 km to ~6 km elevation in the Andes? Some physical explanations and/or corroborating studies of this pattern would be helpful to provide some confidence and context, especially since none of the other SWE datasets have values at these higher elevations on the windward side.

Response: Thank you for your comment on the elevational pattern above 4.5 km in Figure 7. The increase in SWE above 4.5 km is likely due to noises arising from limited number of pixels per elevational bin at high elevation. The total number of pixels above 5 km is only 30% of the pixels number between

4.5-5 km. To avoid the non-representativeness arising from small number of pixels per bin, we used roughly equal number of pixels per elevational bin in the revised Figure 7. The revision only slightly affects absolute values of lapse rate. The key results remain consistent that moderate to high resolution (MR/HR) products show closer snow lapse rate with WUS-SR, whereas low resolution products (LR) underestimate snow lapse rate. Thus, downscaling LR products to high resolution using lapse rate will not resolve the issue of underestimation of snow. We revised the content (in red text) as follows with original text in gay:

"Based on the WUS-SR, climatological $swe_{peak}$ on the windward side of the Sierra Nevada monotonically increases up to ~3.5 km. Across different products, the uncertainty of $swe_{peak}$ is smaller at the lower elevation ~ 1-1.5 km, however, the differences in lapse rate project to larger $swe_{peak}$ uncertainty as elevation increases. The gradients of windward $swe_{peak}$ (i.e., d($swe_{peak}$)/dz) from WUS-SR, averaged over HR and MR products, and averaged over LR products are 0.34 m/km, 0.26 m/km, and 0.05 m/km, respectively. On the leeward side of the Sierra Nevada, the $swe_{peak}$ increases monotonically with elevation from ~ 1 – 3.5 km in the WUS-SR and most of the other products. Similarly, the uncertainty of $swe_{peak}$ is smaller at low elevation from 1 – 2 km and gradually increases with elevation corresponding with the differences in lapse rate across different products. The gradients of leeward $swe_{peak}$ (i.e., d($swe_{peak}$)/dz) from WUS-SR, averaged over HR and MR products, and averaged over LR products are 0.21 m/km, 0.19 m/km, and 0.07 m/km, respectively. HR and MR products have qualitatively similar elevational distributions of $swe_{peak}$ on both the leeward and windward side of the Sierra Nevada for elevations below 3 km, whereas that $swe_{peak}$ from LR are underestimated with large differences in lapse rates compared to WUS-SR.

[Figure]

Figure 7. Elevational distribution of windward and leeward swe_peak in the Sierra Nevada (a) and Andes (b) across reference datasets and products with spatial resolution higher than 1°. Each dot represents the elevation bin-averaged swe_peak.  The number of pixels per bin is roughly equal. GLDAS products at 1° are not included for comparison due to too few points. On the windward side of the subdomains, dots within the red shaded areas are used to compute lapse rates. On the leeward side, dots in the darker shaded areas are used to compute lapse rates.

On the windward side of the Andes, swe_peak from the Andes-SR increases from ~ 1.5 − 3 km, with decreases between 3 and 6 km due to the limitation of moisture. The swe_peak uncertainty is smaller at low elevation bands between ~ 1.5 - 2 km. The uncertainty gets larger as elevation increases from 2 − 3 km corresponding to large differences in positive lapse rates. In contrast, large differences in negative lapse rates above 3 km reduces the uncertainty as elevation increases. The lapse rates of windward swe_peak from the Andes-SR are 0.30 m/km between elevation bands of ~ 1.5 − 3 km and -0.016 m/km between 3 − 6 km (Table S1). On the leeward side, swe_peak increases between ~ 1.5 − 3 km and slightly decrease above 3 km in the Andes-SR. Similar to the windward side, differences in positive lapse rate below 3 km project to larger swe_peak uncertainty as elevation increases from 1.5 km, whereas differences in negative lapse reduces uncertainty as elevation increases above 3 km. The lapse rates of leeward swe_peak from the Andes-SR are 0.22 m/km between elevations of ~ 1.5 − 3 km, and -0.03 m/km between 3 − 6 km.

"